# CONFIDENCE-CALIBRATED ADVERSARIAL TRAINING *and Detection*: MORE ROBUST MODELS GENERALIZING BEYOND THE ATTACK USED DURING TRAINING

## ABSTRACT

Adversarial training is the standard to train models robust against adversarial examples. However, especially for complex datasets, adversarial training incurs a significant loss in accuracy and is known to generalize poorly to stronger attacks, e.g., larger perturbations or other threat models. In this paper, we introduce **confidence-calibrated adversarial training (CCAT)** where the key idea is to enforce that the confidence on adversarial examples decays with their distance to the attacked examples. We show that CCAT preserves better the accuracy of normal training while robustness against adversarial examples is achieved via confidence thresholding, *i.e., detecting adversarial examples based on their confidence*. Most importantly, in strong contrast to adversarial training, the robustness of CCAT generalizes to larger perturbations and other threat models, not encountered during training. *For evaluation, we extend the commonly used robust test error to our detection setting, present an adaptive attack with backtracking and allow the attacker to select, per test example, the* worst-case *adversarial example from multiple black- and white-box attacks. We present experimental results using $L_\infty$, $L_2$, $L_1$ and $L_0$ attacks on MNIST, SVHN and Cifar10.*

## 1 INTRODUCTION

Deep neural networks have shown tremendous improvements in various learning tasks including applications in computer vision, natural language processing or text processing. However, the discovery of adversarial examples, i.e., nearly imperceptibly perturbed inputs that cause mis-classification, has revealed severe security threats, as demonstrated by attacking popular computer vision services such as Google Cloud Vision (Ilyas et al., 2018a) or Clarifai (Liu et al., 2016; Bhagoji et al., 2017b). As the number of safety- and privacy-critical applications is increasing, e.g., autonomous driving or medical imaging, this problem becomes even more important.

In practice, adversarial training and its variants *(Szegedy et al., 2013; Goodfellow et al., 2014; Madry et al., 2018)*, i.e., training on adversarial examples, can be regarded as the state-of-the-art to obtain models robust against adversarial examples. In contrast to many other defenses, and to the best of our knowledge, adversarial training has not been broken so far. However, adversarial training is known to increase test error significantly. Only on simple datasets such as MNIST (LeCun et al., 1998), adversarial training is able to preserve accuracy. This observation is typically described as a trade-off between robustness and accuracy *(Tsipras et al., 2018; Stutz et al., 2019; Raghunathan et al., 2019; Zhang et al., 2019)*. Furthermore, *while adversarial training leads to robust models for the employed threat model used during training, the obtained robustness does not translate to other threat models, e.g., other $L_p$-balls (Sharma & Chen, 2017; Song et al., 2018; Madry et al., 2018; Tramèr & Boneh, 2019; Li et al., 2019a; Kang et al., 2019), larger perturbations than the ones used during training or distal adversarial examples (Hein et al., 2019).*

**Contributions:** We address both problems of adversarial training: *the drop in accuracy on datasets such as Cifar10 (Krizhevsky, 2009) and the poor "generalization" of robustness to larger perturbations or unseen threat models*. To this end, we introduce **confidence-calibrated adversarial training (CCAT)** based on the idea that the predicted confidence for an adversarial example should decrease with its distance to the attacked example. Specifically, we bias the network to predict a

convex combination of uniform and (correct) one-hot distribution for adversarial examples, which becomes uniform as the distance to the attacked test example increases. Thus, CCAT implicitly biases the network to predict the uniform distribution over classes beyond the adversarial examples seen during training. In contrast, standard adversarial training forces the network to classify adversarial examples correctly with maximal confidence but provides no guidance how to extrapolate beyond adversarial examples seen during training. *Overall, CCAT allows to detect previously unseen adversarial examples, e.g., large perturbations, different $L_p$ attacks or distal adversarial examples, by confidence thresholding, i.e., detection.*

*We also discuss a worst-case evaluation methodology using an adaptive attack to compare our detection-based approach with adversarial training. First, we generalize the commonly used robust test error (Madry et al., 2018; Schott et al., 2019) to our detection setting. Second, for each test example, the worst-case adversarial example, i.e., with highest confidence, is selected from multiple white- and black-box attacks. Using an adapted attack based on (Madry et al., 2018; Dong et al., 2018), augmented with a novel backtracking scheme, we show that CCAT produces models with improved accuracy on Cifar10 and robustness to various unseen adversarial examples: larger perturbations, $L_2$, $L_1$ and $L_0$ attacks, distal adversarial examples and corrupted examples on MNIST(-C), (LeCun et al., 1998; Mu & Gilmer, 2019) SVHN (Netzer et al., 2011), Cifar10(-C) (Hendrycks & Dietterich, 2019). We will make our code for CCAT and evaluation publicly available.*

 *Outline:* *After reviewing related work in Sec. 1.1, we discuss adversarial training (Madry et al., 2018) and introduce CCAT in Sec. 2. In Sec. 3 we present our confidence-thresholded robust test error and present experimental results in Sec. 4. Finally, we conclude in Sec. 5.*

## 1.1 RELATED WORK

**Adversarial Examples:** Adversarial examples can roughly be divided into white-box attacks, i.e., with access to the models, its weights and gradients, e.g. (Goodfellow et al., 2014; Madry et al., 2017; Carlini & Wagner, 2017b), and black-box attacks, i.e., only with access to the output of the model, e.g. (Chen et al., 2017; Brendel & Bethge, 2017; Su et al., 2017; Ilyas et al., 2018b; Sarkar et al., 2017; Narodytska & Kasiviswanathan, 2017). *Adversarial examples were also found to be transferable between similar models (Liu et al., 2016; Xie et al., 2018). While these attacks are intended to be (nearly) imperceptible, visible adversarial transformations, e.g., rotations or rotations (Engstrom et al., 2017; 2019; Dumont et al., 2018; Alaifari et al., 2018; Xiao et al., 2018), or patches (Liu et al., 2018; Lee & Kolter, 2019; Brown et al., 2017; Karmon et al., 2018; Zajac et al., 2019) have also been proposed and transferred to the physical world (Athalye et al., 2018b; Li et al., 2019b; Kurakin et al., 2016). We refer to recent surveys (Barreno et al., 2006; Yuan et al., 2017; Akhtar & Mian, 2018; Biggio & Roli, 2018a) for more details..* Recently, white-box attacks utilizing projected gradient ascent to maximize cross-entropy loss or surrogate objectives, e.g., (Madry et al., 2017; Dong et al., 2018; Carlini & Wagner, 2017b), have become standard. Instead, we directly maximize the confidence in any but the true class, similar to (Hein et al., 2019; Goodfellow et al., 2019), to attack our proposed training procedure, CCAT.

**Adversarial Training:** Many defenses against adversarial attacks have been proposed, see, e.g., (Yuan et al., 2017; Akhtar & Mian, 2018; Biggio & Roli, 2018b), of which some have been shown to be ineffective, e.g., in (Athalye et al., 2018a; Athalye & Carlini, 2018). *Currently, adversarial training, i.e., training on adversarial examples, is the de-facto standard to obtain robust models.* While adversarial training was proposed in different variants (Szegedy et al., 2013; Zantedeschi et al., 2017; Miyato et al., 2016; Huang et al., 2015; Shaham et al., 2018; Sinha et al., 2018; Madry et al., 2017; Wang et al., 2019), the formulation by Madry et al. (2017) received considerable attention and has been extended in various ways (Lee et al., 2017; Ye & Zhu, 2018; Liu et al., 2019). For example, in (Shafahi et al., 2018; Pérolat et al., 2018), adversarial training is applied to universal adversarial example, in (Cai et al., 2018), curriculum learning is used, and in (Tramèr et al., 2017; Grefenstette et al., 2018) ensemble adversarial training is proposed. *Adversarial training is also used in provable/certified defenses (Kolter & Wong, 2017; Zhang & Evans, 2018). The increased sample complexity of adversarial training (Schmidt et al., 2018) has been addressed in (Lamb et al., 2019) by training on interpolated examples or in (Carmon et al., 2019; Uesato et al., 2019) using unlabeled examples. The dependence on the threat model used during training was addressed in (Tramèr & Boneh, 2019) by using multiple threat models during training. Finally, the frequently observed trade-off between accuracy and robustness has been studied both theoretically and empir-*

*ically (Tsipras et al., 2018; Stutz et al., 2019; Zhang et al., 2019; Raghunathan et al., 2019).* CCAT differs from standard adversarial training in the used attack objective and the imposed distribution over the labels, which tends towards a uniform distribution for large perturbations.

***Detection:*** *Instead of correctly classifying adversarial examples, as intended in adversarial training, several works (Gong et al., 2017; Rouhani et al., 2018; Grosse et al., 2017; Feinman et al., 2017; Liao et al., 2018; Ma et al., 2018; Amsaleg et al., 2017; Metzen et al., 2017; Bhagoji et al., 2017a; Hendrycks & Gimpel, 2017; Li & Li, 2017; Pang et al., 2018) try instead to detect adversarial examples. However, several detectors have been shown to be ineffective against adaptive attacks, i.e., adversaries aware of the used detection mechanism (Carlini & Wagner, 2017a). Recently, the detection of adversarial examples based on their confidence, similar to our approach with CCAT, has also been discussed (Pang et al., 2018); however, the authors build on (Feinman et al., 2017) which has been shown to be ineffective (Carlini & Wagner, 2017a), as well. Goodfellow et al. (2019), instead, focus on evaluating confidence-based detection methods using adaptive, targeted attacks maximizing confidence. Our attack, introduced in Sec. 2.2, is similar in spirit, however, untargeted by nature and, thus, suitable for usage with CCAT.*

## 2    CONFIDENCE CALIBRATION OF ADVERSARIAL EXAMPLES

Adversarial training, as described by Madry et al. (2018), has become standard to train robust models. However, accuracy on challenging datasets such as Cifar10 is reduced and robustness does not generalize to larger perturbations, e.g., with respect to the used $\mathcal{L}_\infty$ constraint, *or different threat models in other $L_p$ norms. With CCAT, we intend to address both shortcomings: Instead of training models to* correctly classify *adversarial examples, we want to* detect *adversarial examples based on their confidence. In the following, we first review adversarial training, before discussing the modifications needed for our CCAT, thereby turning robustness against adversarial examples into a detection problem.*

**Notation:** We consider a classifier $f : \mathbb{R}^d \rightarrow \mathbb{R}^K$ where $K$ is the number of classes and $f_k$ denotes the confidence for class $k$. While we assume that the cross-entropy loss $\mathcal{L}$ is used for training, our approach can also be used with other losses. Given $x \in \mathbb{R}^d$, classified correctly as $y = \operatorname{argmax}_k f_k(x)$, an *adversarial example* $x + \delta$ is defined as a "small" *perturbation* $\delta$ such that $\operatorname{argmax}_k f_k(x + \delta) \neq y$, i.e., the classifier changes its decision. The strength of the change $\delta$ is measured by some $L_p$-*norm* with $p \in \{0, 1, 2, \infty\}$; $p = \infty$ is a popular choice in the literature as this leads to the smallest perturbation per feature/pixel.

### 2.1    ADVERSARIAL TRAINING

Standard adversarial training, as formulated by Madry et al. (2018), is given as the following min-max problem:

$$\min_{w} \mathbb{E} \left[ \max_{\|\delta\|_\infty \leq \epsilon} \mathcal{L}(f(x + \delta; w), y) \right] \tag{1}$$

with $w$ being the classifier's parameters and $\mathcal{L}$ being the cross-entropy loss. During mini-batch training the inner maximization problem,

$$\max_{\|\delta\|_\infty \leq \epsilon} \mathcal{L}(f(x + \delta; w), y), \tag{2}$$

is approximately solved. In addition to the $L_\infty$-constraint, a box constraint, i.e., $\tilde{x}_i = (x + \delta)_i \in [0, 1]$, is enforced for images. Note that maximizing the cross-entropy loss is equivalent to finding the adversarial example with *minimal* confidence in the true class. For neural networks, this is generally a non-convex optimization problem. In (Madry et al., 2018) the problem is tackled using projected gradient descent (PGD), which is typically initialized using a random $\delta$ with $\|\delta\|_\infty \leq \epsilon$. *During adversarial training, PGD is applied to the batch of every iteration in order to update the classifier's parameters on the obtained adversarial examples.* At test time one uses the best out of several random restarts to assess robustness.

In contrast to adversarial training as proposed in (Madry et al., 2018), which computes adversarial examples for the *full* batch in each iteration, others compute adversarial examples only for half the

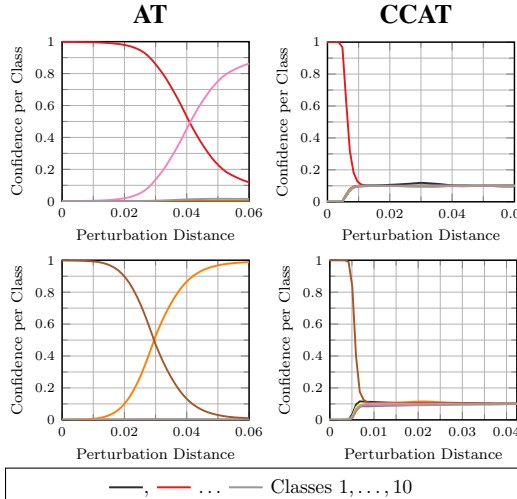

Figure 1: **Confidence Calibration.** For adversarial training (AT) and our confidence-calibrated adversarial training (CCAT) with $\rho_{\text{pow}} = 10$ using the power transition in Eq. (6), we plot the probabilities for all ten classes along adversarial directions. *Adversarial examples were computed using our $L_\infty$-PGD-Conf attack, cf. Sec. 4.1: using projected gradient descent (Madry et al., 2018) the confidence of the adversarial examples is maximized for $T = 1000$ and $L_\infty$-constraint $\epsilon = 0.03$.* The robustness of AT does not generalize beyond the $\epsilon = 0.03$-ball as high confidence adversarial examples can be found for larger perturbations, whereas CCAT predicts close to uniform confidence after some transition phase allowing to easily detect adversarial examples.

examples of each batch (Szegedy et al., 2013), *i.e., instead of training* only *on adversarial examples, each batch is divided* 50% *into clean and* 50% *into adversarial examples*. Compared to Eq. (1), 50%/50% *adversarial training* effectively minimizes

$$\min_w \underbrace{\mathbb{E}\Big[\max_{\|\delta\|_\infty \leq \epsilon} \mathcal{L}(f(x + \delta; w), y)\Big]}_{\text{50\% adversarial training}} + \underbrace{\mathbb{E}\big[\mathcal{L}(f(x; w), y)\big]}_{\text{50\% "clean" training}} \tag{3}$$

This improves test accuracy on clean examples *compared to* 100% *adversarial training*, i.e., Eq. (1), but typically leads to worse robustness. Intuitively, *by balancing both terms in Eq. (3), the trade-off between accuracy and robustness can already be optimized to some extent (Stutz et al., 2019).*

There are two problems with adversarial training: First, the $\epsilon$-ball around training examples might include examples from other classes. Then, the attack, i.e., the inner maximization in Eq. (1), will focus on these regions such that adversarial training for these examples gets "stuck". This case is illustrated in our theoretical toy dataset in Sec. 2.3 *and has also been considered in related work (Jacobsen et al., 2019b;a).* Here, both 100% and 50%/50% adversarial training, cf. Eq. (1) and (3), are not able to find the Bayes optimal classifier in a fully deterministic problem, i.e., zero Bayes error. This might contribute to the observed drop in accuracy for adversarial training on, e.g., Cifar10 (Krizhevsky, 2009). Second, and more importantly, adversarial training as in Eq. (1) does not guide the classifier how to extrapolate beyond the used $\epsilon$-ball during training. Even worse, it enforces high confidence predictions within the $\epsilon$-ball, which clearly cannot be extrapolated to arbitrary regions. It is not surprising that adversarial examples can often be found right beyond the $\epsilon$-ball, i.e., using larger perturbations than at training time or other threat models, e.g., other $L_p$ norm balls.

## 2.2 CONFIDENCE-CALIBRATED ADVERSARIAL TRAINING

*Addressing these problems requires only few but effective modifications as outlined in the description of our proposed **confidence-calibrated adversarial training (CCAT)** in Alg. 1: during training, we bias the network to predict a uniform distribution over the classes on adversarial examples that are sufficiently far away from the original training examples. Subsequently, during testing, adversarial examples can be detected by confidence thresholding – adversarial examples receive near-uniform confidence while test examples receive high-confidence. Thus, given an example $x$, the ideal attacker during training* maximizes the confidence in any arbitrary other label $k \neq y$ *instead of minimizing the confidence in the true label $y$, as in Eq. (1):*

$$\max_{\|\delta\|_\infty \leq \epsilon} \max_{k \neq y} f_k(x + \delta; w) \tag{4}$$

*where $f_k$ denotes the probability $f$ assigns to class $k$ (i.e., in practice, $f$ is the output of the softmax layer).* A similar objective but for targeted attacks has been used in (Goodfellow et al., 2019), whereas our goal is an untargeted attack and thus our objective is the maximal confidence over all

**Algorithm 1 Pseudo-code of confidence-calibrated adversarial training (CCAT).** The main changes compared to regular adversarial training as, e.g., described in (Madry et al., 2018) or (Szegedy et al., 2013), are in the attack (line 4) and the probability distribution over the classes (line 6,7), which becomes more uniform as distance $\|\delta\|_\infty$ increases.

1: **while** true **do**
2:     choose random batch $(x_1, y_1), \ldots, (x_B, y_B)$.
3:     **for** $b = 1, \ldots, B/2$ **do**
4:         {maximize confidence in other classes than true one for adversarial example $\tilde{x}_b$, *Eq. (4)*:}
5:         $\delta_b := \text{argmax}_{\|\delta\|_\infty \leq \epsilon} \max_{k \neq y_b} f_k(x_b + \delta)$
6:         $\tilde{x}_b := x_b + \delta_b$
7:         {probability over classes of $\tilde{x}_b$ becomes more uniform as $\|\delta_b\|_\infty$ increases, *Eq. (6)*:}
8:         $\lambda := e^{-\rho\|\delta_b\|_\infty}$ or $\lambda := (1 - \min(1, \|\delta\|_\infty/\epsilon))^\rho$
9:         {$\tilde{y}_b$ is convex combination of one hot and uniform distribution over the classes, *Eq. (5)*:}
10:        $\tilde{y}_b := \lambda \, \text{one\_hot}(y_b) + \frac{(1-\lambda)}{K} \mathbb{1}$
11:     **end for**
12:     *{corresponds to 50%/50% adversarial training, Eq. (3):}*
13:     update parameters using $\sum_{b=1}^{B/2} \mathcal{L}(f(\tilde{x}_b), \tilde{y}_b) + \sum_{b=B/2}^{B} \mathcal{L}(f(x_b), y_b)$
14: **end while**

other classes. Then, during training, CCAT biases the classifier towards predicting uniform distributions on adversarial examples by using the following distribution as target in the cross-entropy loss:

$$\hat{p}(k) = \lambda p_y(k) + (1 - \lambda)u(k) \quad k = 1, \ldots, K. \tag{5}$$

Here, $p_y(k)$ is the original "one-hot" distribution, i.e., $p_y(k) = 1$ iff $k = y$ and $p_y(k) = 0$ otherwise with $y$ being the true label, and $u(k) = 1/K$ is the uniform distribution. Thus, we enforce a convex combination of the original label distribution and the uniform distribution which is controlled by the parameter $\lambda$. We choose $\lambda$ to decrease with the distance $\|\delta\|_\infty$ of the adversarial example to the attacked example $x$ *with the intention to enforce uniform predictions when $\|\delta\|_\infty = \delta$. Then, the network is encouraged to extrapolate this uniform distribution beyond the used $\epsilon$-ball. Even if extrapolation does not work perfectly, the uniform distribution is much more meaningful for extrapolation in the region between classes compared to high-confidence predictions as encouraged in standard adversarial training.* We consider two variants of transitions, i.e., controlling the trade-off $\lambda$ between one-hot and uniform distribution:

$$\begin{aligned} \lambda &= e^{-\rho\|\delta\|_\infty} & \text{(``exponential transition'' (exp))} \\ \lambda &= (1 - \min(1, \|\delta\|_\infty/\epsilon))^\rho & \text{(``power transition'' (pow))} \end{aligned} \tag{6}$$

This ensures that for $\delta = 0$ we impose the original (one-hot) label. For growing $\delta$, however, the influence of the original label decays proportional to $\|\delta\|_\infty$. The speed of decay is controlled by the parameter $\rho$. For the exponential transition, we always have a bias towards the true label as even for large $\rho$, $\lambda$ will be non-zero. In case of the power transition, $\lambda = 0$ for $\|\delta\|_\infty \geq \epsilon$, meaning a pure uniform distribution is enforced. It is important to note that in CCAT in Alg. 1 we train on 50% clean and 50% adversarial examples in each batch, *as in Eq. (3)*. Training only on adversarial examples will not work as we the network has no incentive to predict correct labels.

## 2.3   Confidence-Calibrated Adversarial Training Yields Accurate Models

Proposition 1 analyzes 100% adversarial training and its 50%/50% variant as well as our confidence-calibrated variant, CCAT, *to show that there exist problems where both 100% and 50%/50% adversarial training are unable to reconcile robustness and accuracy, as recently discussed (Tsipras et al., 2018; Stutz et al., 2019; Raghunathan et al., 2019; Zhang et al., 2019).* However, our CCAT is able to obtain *both* robustness and accuracy given that $\lambda$ in Eq. (6) is chosen appropriately.

**Proposition 1.** *We consider a classification problem with two points $x = 0$ and $x = \epsilon$ in $\mathbb{R}$ with deterministic labels, that is $p(y = 2|x = 0) = 1$ and $p(y = 1|x = \epsilon) = 1$ and the problem is fully determined by the probability $p_0 = p(x = 0)$ as $p(x = \epsilon) = 1 - p_0$. The Bayes error of this*

*classification problem is zero. The predicted probability distribution over the classes is $\tilde{p}(y|x) = \frac{e^{g_y(x)}}{e^{g_1(x)}+e^{g_2(x)}}$, where $g : \mathbb{R}^d \to \mathbb{R}^2$ and we assume that the function $\lambda : \mathbb{R}_+ \to [0,1]$ is monotonically decreasing and $\lambda(0) = 1$. The Bayes optimal classifier for the cross-entropy loss of*

- *adversarial training on $100\%$ adversarial examples, cf. Eq. (1), yields an error of $\min\{p_0, 1 - p_0\}$.*

- *adversarial training with $50\%$ adversarial and $50\%$ clean examples per batch, cf. Eq. (3), yields an error of $\min\{p_0, 1 - p_0\}$.*

- *CCAT on $50\%$ clean and $50\%$ adversarial examples, cf. Alg. 1, yields zero error if $\lambda(\epsilon) < \min\left\{\frac{p_0}{1-p_0}, \frac{1-p_0}{p_0}\right\}$.*

## 3 CONFIDENCE-THRESHOLDED ROBUST TEST ERROR (RERR)

*CCAT runs in a two-stage process: First, examples are rejected by confidence thresholding. Ideally, all adversarial examples are rejected as CCAT encourages low-confidence adversarial examples. Second, we evaluate robustness and accuracy on the non-rejected (i.e., correctly or incorrectly classified) examples. In the first stage (i.e., confidence thresholding), we consider* successful *adversarial examples as negatives and* correctly classified *test examples as positives. Then, we report the area under the ROC curve, i.e., ROC AUC, which shows how well we can discriminate adversarial examples from correctly classified clean examples. In the second stage (i.e., after detection by confidence thresholding), we fix the threshold $\tau$ at a true positive rate (TPR) of $99\%$ for correctly classified test samples, i.e., the network is allowed to reject at most $1\%$ of correctly classified test examples. We note that this also improves accuracy as errors on test examples typically have lower confidence. Then, we intend to use test error (Err) and robust test error (RErr), as also used in (Madry et al., 2018), to evaluate adversarial training. However, extending RErr to take into account confidence-thresholding as in our detection setting is non-trivial: for example, correctly classified examples can can have lower confidence than their corresponding adversarial examples.* Thus, given confidence threshold $\tau$, we define the confidence-thresholded robust test error RErr as follows:

$$\mathrm{RErr}(\tau) = \frac{\sum_{n=1}^{N} \mathbb{1}_{f(x_n) \neq y_n} \mathbb{1}_{c(x_n) \geq \tau} + \sum_{n=1}^{N} \mathbb{1}_{f(x_n) = y_n} \mathbb{1}_{f(\tilde{x}_n) \neq y_n} \mathbb{1}_{c(\tilde{x}_n) \geq \tau}}{\sum_{n=1}^{N} \mathbb{1}_{c(x_n) \geq \tau} + \sum_{n=1}^{N} \mathbb{1}_{c(x_n) < \tau} \mathbb{1}_{c(\tilde{x}_n) \geq \tau} \mathbb{1}_{f(x_n) = y_n} \mathbb{1}_{f(\tilde{x}_n) \neq y_n}}. \tag{7}$$

Here, $\tau$ is the confidence-threshold fixed on a held-out validation set, $\{(x_n, y_n)\}_{n=1}^{N}$ are test examples, $c(x_n) := \max_k f_k(x_n)$ *denotes the classifier's confidence on $x_n$*, $f(x_n) := \arg\max_k f_k(x_n)$ *denotes the classifier's decision*, and $\tilde{x}_n$ are adversarial examples. The numerator counts the number of incorrectly classified test examples $x_n$ with $c(x_n) \geq \tau$ (first term) and the number of *successful* adversarial examples $\tilde{x}_n$ on *correctly classified* test examples with $c(\tilde{x}_n) \geq \tau$ (second term). The denominator counts test examples $x_n$ with $c(x_n) \geq \tau$ (first term) and the number of *successful* adversarial examples $\tilde{x}_n$ with $c(\tilde{x}_n) \geq \tau$ but where the corresponding test example $x_n$ has $c(x_n) < \tau$ (second term). The latter takes care of the special case where adversarial examples have higher confidence than their corresponding test examples, *as mentioned above and* is encouraged by the objective of our attack in Eq. (4). In total this yields a correct fraction within $[0, 1]$ and for $\tau = 0$, Eq. (7) reduces to the "standard" RErr. The confidence-thresholded $\mathrm{Err}(\tau)$ corresponds to taking only the first terms in both numerator and denominator.

## 4 EXPERIMENTS

We compare normal training, adversarial training (AT) and our CCAT on MNIST, (LeCun et al., 1998) SVHN (Netzer et al., 2011) and Cifar10 (Krizhevsky, 2009). *We use ResNet-20 (He et al., 2016), implemented in PyTorch (Paszke et al., 2017), initialized following (He et al., 2015) and trained using stochastic gradient descent with batch size of $100$ for $100$ or $200$ epochs (MNIST and SVHN/Cifar10, respectively). As detailed in Sec. 3, we report* **ROC AUC (higher is better)** *as well as our confidence-thresholded* **test error (Err; lower is better)** *and* **robust test error (RErr; lower is better)**; *Err is reported on the full test sets, while ROC AUC and RErr are reported on the first $1000$ attacked test examples. More details and experimental results can be found in Appendix B.*

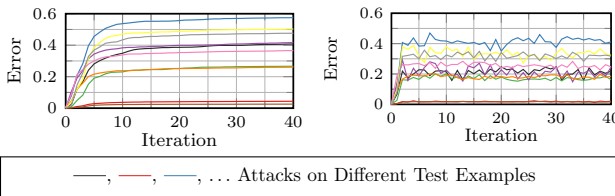

Figure 2: **Momentum and backtracking.** Our $L_\infty$ PGD-Conf attack, *cf. Sec. 4.1*, with 40 iterations with momentum and backtracking (left) and without both (right). We plot the objective of Eq. (4) over iterations for 10 samples (different colors).

## 4.1 ATTACKS

*White-box and adaptive attacks:* We follow (Madry et al., 2018) and use projected gradient descent (PGD) to minimize the negatives of Eq. (2) and (4); we denote them as PGD-CE and PGD-Conf. The perturbation $\delta$ is initialized uniformly over direction and distance; for PGD-Conf, we additionally use $\delta = 0$ as initialization. Different from (Madry et al., 2018), we run exactly $T$ iterations (no early stopping) and take the perturbation corresponding to the best objective of the $T$ iterations. In addition to momentum, as in (Dong et al., 2018), we propose to use an adaptive learning rate in combination with a backtracking scheme to improve the attacks: after each iteration, the computed update is only applied if it improves the objective; otherwise the learning rate is reduced. *We also implemented PGD for $L_2$, $L_1$ and $L_0$ attacks; note that for each $L_p$, $p \in \{\infty, 2, 1, 0\}$, PGD-Conf has been explicitly designed to attack our CCAT. We train on $L_\infty$ attacks using $T = 40$ iterations with $\epsilon = 0.3$ (MNIST) or $\epsilon = 0.03$ (SVHN/Cifar10).* For evaluation, we use $T = 2000$ iterations and 10 random retries for PGD-Conf; $T = 200$ with 50 random retries for PGD-CE. *For $L_2$, $L_1$ and $L_0$ attacks, we set $\epsilon$ to $3, 10, 15$ (MNIST) or $1, 7.85, 10$ (SVHN/Cifar10), respectively, partly following (Tramèr & Boneh, 2019). Learning rates have been tuned for each $L_p$ and objective (i.e., CE or Conf) independently.*

*Black-box attacks:* For the $L_\infty$ case, we additionally use random sampling, the attack by Ilyas et al. (2018a), adapted with momentum and backtracking optimizing Eq. (4) for $T = 2000$ iterations with 10 attempts, a variant of (Narodytska & Kasiviswanathan, 2017) with $T = 2000$ iterations and the "cube" attack (Andriushchenko, 2019) with $T = 5000$ iterations. *We also consider the cube attack in the $L_2$ setting and the attack of (Croce & Hein, 2019) in the $L_0$ setting.* The black-box attacks ensure that our defense avoids, e.g., gradient masking as described in (Athalye et al., 2018a).

## 4.2 WORST-CASE EVALUATION METHODOLOGY

*Instead of reporting robustness against individual attacks, as commonly done in the literature, we use a* worst-case *evaluation scheme: For each individual test example, all adversarial examples from multiple attempts (i.e., multiple random restarts) and across all (white- and black-box) attacks are accumulated. Subsequently, per text example, only the adversarial example with highest confidence is kept; these are the worst-case adversarial examples for each test example. In practice, this scheme allows to accumulate results from different attacks and thus corresponds to a much stronger robustness evaluation than usual.*

| **SVHN:** Attack Ablation with RErr for $\tau$@99%TPR ($L_\infty$ attack with $\epsilon = 0.03$) | | | | | | | |
|---|---|---|---|---|---|---|---|
| Optimization | momentum+backtrack | | | | | mom | − |
| Initialization | zero | | | | rand | zero | zero |
| Iterations $T$ | 40 | 200 | 2000 | 4000 | 4000 | 300 | 300 |
| AT | 38.4 | 46.2 | 49.9 | 50.1 | 51.8 | 38.1 | 30.8 |
| AT Conf | 27.4 | 40.5 | 46.9 | 47.3 | 48.1 | 28.5 | 23.8 |
| CCAT | 4.0 | 5.0 | 22.8 | 23.3 | 5.2 | 2.6 | 2.6 |

Table 1: **Attack ablation study on SVHN.** Comparison of our $L_\infty$ PGD-Conf attack with $\epsilon = 0.03$ on the test set for different number of iterations $T$ and configurations of momentum, backtracking and initialization. As backtracking needs an additional forward pass per iteration, we compare $T = 200$ with backtracking to $T = 300$ without. Attacks on adversarial training (AT) succeed within a few iterations, but are more difficult against CCAT and require initialization at zero.

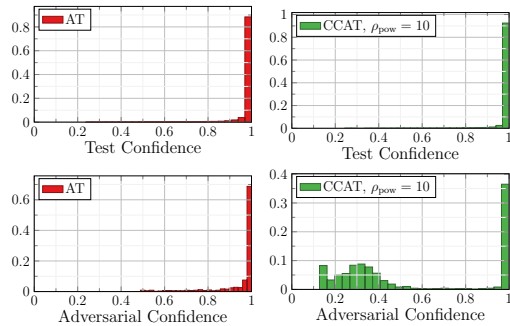

Figure 3: **Confidence histograms on SVHN.** For AT (left) and CCAT (right) with $\rho_{\text{pow}} = 10$ (right), we show confidence histograms corresponding to *correctly classified* test examples (top) and *successful* adversarial examples (bottom). We consider the worst-case adversarial examples across all tested $L_\infty$ attacks for $\epsilon = 0.03$.

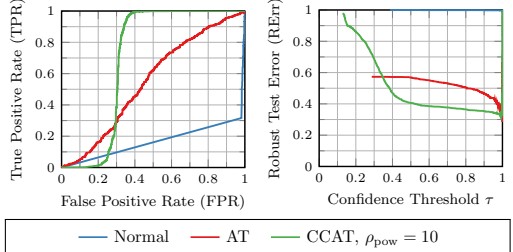

Figure 4: **ROC and RErr curves on SVHN.** Left: ROC curves, i.e., FPR against TPR when distinguishing *correctly classified* test examples from *successful* adversarial examples by confidence. Right: RErr against confidence threshold $\tau$. For evaluation, we choose $\tau$ in order to obtain 99%TPR. As described in the text, RErr subsumes both Err and FPR. Curves based on worst-case examples across all tested $L_\infty$ attacks.

### 4.3 ABLATION STUDY

**Momentum and Backtracking:** Fig. 2 illustrates the advantage of momentum and the proposed backtracking scheme for PGD-Conf with $T = 40$ iterations on 10 test examples of SVHN. As shown in Fig. 2 and Tab. 1, better objective values can be achieved within fewer iterations and avoiding oscillation which is important at training time. However, also at test time, Tab. 1 shows that attacking our CCAT model effectively requires up to $T = 2000$ iterations and zero initialization so that RErr for $\tau$@99%TPR stagnates. In contrast, PGD-Conf performs better against AT even for smaller $T$ and without momentum or backtracking. Thus, finding high-confidence adversarial examples against CCAT is more difficult than for AT. Overall, this illustrates our immense effort put into attacking our proposed defense with an adapted attack, *novel optimization techniques, and large number of iterations.*

**Evaluation Metrics:** Fig. 4 shows ROC and RErr curves on SVHN, considering AT and CCAT with $\rho_{\text{pow}} = 10$, i.e., using the power transition from Eq. (6). The ROC curves, and the corresponding AUC value, quantify how well (successful) adversarial examples can be *detected, i.e., distinguished* from (correctly classified) test examples. Note our conservative choice to use the confidence threshold $\tau$ at 99%TPR, wrongly rejecting at most 1% correctly classified examples. We note that $\tau$ depends only on correctly classified test examples, not adversarial examples, and RErr implicitly includes FPR as well as Err. *We focus on the confidence-thresholded variants of RErr and Err as confidence thresholding will naturally also improve results for AT, resulting in a fair comparison, i.e., AT is also allowed to benefit from our two-stage detection setting; for Err this can be seen in Tab. 2.* On SVHN and Cifar10, we found the power transition with $\rho_{\text{pow}} = 10$ from Eq. (6) to work best. Up to $\rho_{\text{pow}} = 10$, performance regarding RErr for $\tau$@99% TPR continuously improves and after $\rho_{\text{pow}} = 10$ performance stagnates, as shown in detail in the appendix. On MNIST, interestingly, exponential transition with $\rho_{\text{exp}} = 7$ performs best *against $L_\infty$ attacks*; we assume that the

| | MNIST: Test Error | | SVHN: Test Error | | SVHN: Test Error | |
|---|---|---|---|---|---|---|
| | **Standard** | **Detection** | **Standard** | **Detection** | **Standard** | **Detection** |
| | $\tau = 0$ | $\tau$@99%TPR | $\tau = 0$ | $\tau$@99%TPR | $\tau = 0$ | $\tau$@99%TPR |
| | Err in % | Err in % | Err in % | Err in % | Err in % | Err in % |
| Normal | 0.4 | 0.1 | 3.6 | 2.6 | **8.3** | 7.4 |
| AT | 0.5 | **0** | 3.4 | 2.5 | 16.6 | 15.5 |
| CCAT | **0.3 (0.3)** | 0.1 (0.1) | **2.9** | **2.1** | 10.1 | **6.7** |

Table 2: *Test errors on MNIST, SVHN and Cifar10. Err for $\tau = 0$, i.e., standard evaluation, and $\tau$@99%TPR, i.e., detection evaluation, comparing normal training, AT and CCAT. Especially on Cifar10, AT increases Err significantly, in both settings. CCAT, in contrast, is able to preserve the Err of the normally trained model better.*

| | $L_\infty, \epsilon = 0.30$ | | $L_\infty, \epsilon = 0.40$ | | $L_2, \epsilon = 3$ | | $L_1, \epsilon = 10.00$ | | $L_0, \epsilon = 15$ | |
|---|---|---|---|---|---|---|---|---|---|---|
| **MNIST:** Main Results for **Detection** $\tau$@99%TPR $(L_\infty, \epsilon = 0.30$ during training, standard evaluation, i.e., $\tau = 0$, in Appendix B) | | | | | | | | | | |
| | (seen) | | (unseen) | | (unseen) | | (unseen) | | (unseen) | |
| | ROC AUC | RErr in % | ROC AUC | RErr in % | ROC AUC | RErr in % | ROC AUC | RErr in % | ROC AUC | RErr in % |
| AT | 0.97 | **1.0** | 0.20 | 100.0 | 0.73 | 81.3 | 0.93 | 5.2 | **0.99** | **2.5** |
| CCAT, $\rho_{\mathrm{exp}}{=}7$ | **0.99** | 7.7 | **0.94** | 40.0 | **1.00** | 1.4 | 0.96 | 14.7 | **0.99** | 7.8 |
| CCAT, $\rho_{\mathrm{pow}}{=}10$ | 0.96 | 21.9 | **0.94** | 29.0 | **1.00** | **0.1** | **1.00** | **1.6** | **0.99** | 7.8 |

| | $L_\infty, \epsilon = 0.03$ | | $L_\infty, \epsilon = 0.06$ | | $L_2, \epsilon = 1$ | | $L_1, \epsilon = 7.85$ | | $L_0, \epsilon = 10$ | |
|---|---|---|---|---|---|---|---|---|---|---|
| **SVHN:** Main Results for **Detection** $\tau$@99%TPR $(L_\infty, \epsilon = 0.03$ during training, standard evaluation, i.e., $\tau = 0$, in Appendix B) | | | | | | | | | | |
| | (seen) | | (unseen) | | (unseen) | | (unseen) | | (unseen) | |
| | ROC AUC | RErr in % | ROC AUC | RErr in % | ROC AUC | RErr in % | ROC AUC | RErr in % | ROC AUC | RErr in % |
| AT | 0.55 | 55.6 | 0.32 | 88.3 | 0.26 | 92.0 | 0.34 | 91.9 | 0.90 | 73.4 |
| CCAT | **0.70** | **38.5** | **0.70** | **46.0** | **0.91** | **18.5** | **0.89** | **20.8** | **1.00** | **2.7** |

| | $L_\infty, \epsilon = 0.03$ | | $L_\infty, \epsilon = 0.06$ | | $L_2, \epsilon = 1$ | | $L_1, \epsilon = 7.85$ | | $L_0, \epsilon = 10$ | |
|---|---|---|---|---|---|---|---|---|---|---|
| **CIFAR10:** Main Results for **Detection** $\tau$@99%TPR $(L_\infty, \epsilon = 0.03$ during training, standard evaluation, i.e., $\tau = 0$, in Appendix B) | | | | | | | | | | |
| | (seen) | | (unseen) | | (unseen) | | (unseen) | | (unseen) | |
| | ROC AUC | RErr in % | ROC AUC | RErr in % | ROC AUC | RErr in % | ROC AUC | RErr in % | ROC AUC | RErr in % |
| AT | **0.64** | **62.3** | 0.35 | 93.6 | 0.59 | 73.9 | 0.61 | 68.0 | 0.80 | 74.1 |
| CCAT | 0.60 | 67.9 | **0.43** | **91.5** | **0.77** | **46.2** | **0.78** | **45.2** | **0.98** | **20.9** |

Table 3: ***Main results on MNIST, SVHN and Cifar10.*** Comparison of AT and CCAT on MNIST (top), SVHN (middle) and Cifar10 (bottom). *Per threat model, i.e., for $L_\infty$, $L_2$, $L_1$ and $L_0$ attacks*, we report worst-case results across all tested attacks; the used $\epsilon$ values are reported in the corresponding columns. During training, $L_\infty$ attacks with $\epsilon = 0.3$ on MNIST and $\epsilon = 0.03$ on SVHN/Cifar10 were used; *adversarial examples from the remaining threat models were not encountered during training. We report ROC AUC as well as confidence-thresholded Err and RErr for $\tau$@99%TPR. CCAT is competitive with AT on the $L_\infty$ attacks seen during training, but generalizes significantly better to previously unseen attacks.*

slight bias towards the true label preserved in the exponential transition helps. *However, even on MNIST, the power transition improves robustness against previously unseen attacks, e.g., $L_2$, $L_1$ or $L_0$ attacks, such that we include results for both.*

### 4.4 RESULTS

In Tab. 3, we report the main results of our paper, namely robustness across all evaluated $L_\infty$, $L_2$, $L_1$ and $L_0$ attacks; for $L_\infty$ we include results for the $\epsilon$ used during training and an increased $\epsilon$. While on MNIST, CCAT incurs a drop of roughly 6% in RErr, and on Cifar10 a drop of roughly 5%, both against $L_\infty$ with the same $\epsilon$ as during training, it significantly outperforms AT on SVHN, by more than 16%. On SVHN and Cifar10, Err is additionally improved – on Cifar10, the improvement is particularly significant with roughly 6%. For $L_\infty$ attacks with larger $\epsilon$, robustness of AT degrades significantly, while CCAT is able to preserve robustness to some extent, especially on SVHN. Only on Cifar10, RErr degrades similarly to AT. In terms of generalization to previously unseen threat models, i.e., $L_2$, $L_1$ and $L_0$ attacks, CCAT outperforms AT significantly. *We note that on all datasets, the considered $\epsilon$-balls for $L_2$, $L_1$ and $L_0$ are* not *contained in the $L_\infty$-ball used during training. Overall, robustness of AT degrades significantly under threat models not seen during training, while CCAT generalizes to new threat models much better.*

In Tab. 4 we also report results for detecting uniform noise and adversarial uniform noise (i.e., distal adversarial examples) based on their confidence. For the latter, we sample uniform noise and subsequently use PGD-Conf to maximize the confidence (without considering any true label in Eq. (4)) in the $L_\infty$-ball around the noise point. Although ROC AUC values are high on uniform noise, an FPR of 40% or higher shows that normal training and AT assigns high confidence to uniform noise. When

| | | MNIST ($\epsilon = 0.3$) | | SVHN ($\epsilon = 0.03$) | | CIFAR10 ($\epsilon = 0.03$) | |
|---|---|---|---|---|---|---|---|
| | | **Detection** $\tau$@99%TPR | | **Detection** $\tau$@99%TPR | | **Detection** $\tau$@99%TPR | |
| | | ROC AUC | FPR in % | ROC AUC | FPR in % | ROC AUC | FPR in % |
| Rand | Normal | 0.34 | 100.0 | 0.95 | 72.8 | 0.83 | 87.1 |
| | AT | 0.99 | 44.5 | **1.00** | 0.9 | 0.60 | 100.0 |
| | CCAT | **1.00** | **0.0** | **1.00** | **0.0** | **1.00** | **0.0** |
| Dist | Normal | 0.34 | 100.0 | 0.43 | 100.0 | 0.49 | 100.0 |
| | AT | 0.63 | 100.0 | 0.89 | 98.8 | 0.31 | 100.0 |
| | CCAT | **1.00** | **0.0** | **1.00** | **0.0** | **1.00** | **0.0** |

Table 4: **Results for noise and distal adversarial examples on MNIST, SVHN and Cifar10.** Robustness against uniform noise ("Rand") and distal adversarial examples ("Dist"), i.e., high-confidence adversarial computed on uniform noise using our $L_\infty$ PGD-Conf attack by considering Eq. (4) without any true label; we use $\epsilon = 0.3$ on MNIST, $\epsilon = 0.03$ on SVHN/Cifar10. We report ROC AUC and FPR for a confidence threshold of $\tau$@99%TPR. *In contrast to normal training and AT, CCAT is able to reliably detect all (adversarial) noise examples.*

maximizing confidence on uniform noise, FPR approaches $100\%$ on all datasets. In contrast CCAT allows to separate these attacks perfectly from test examples. *Furthermore, Tab. 5 presents results on MNIST-C and Cifar10-C (Mu & Gilmer, 2019; Hendrycks & Dietterich, 2019), containing the MNIST and Cifar10 test examples with various corruptions (e.g., spatial transformations, brightness/contrast changes, noise, etc.). Again, we consider our two-stage detection approach, reporting ROC AUC for detecting corrupted examples based on confidence and confidence-thresholded Err* on the corrupted examples. *The results are averaged over all available corruptions (15 on MNIST, 19 on Cifar10). As can be seen, allowing confidence thresholding is beneficial in this scenario for AT and CCAT. However, CCAT, outperforms normal training and AT significantly.*

## 5 CONCLUSION

We proposed **confidence-calibrated adversarial training (CCAT)** which addresses two problems of standard adversarial training: an apparent accuracy-robustness problem, i.e., adversarial training tends to worsen accuracy *on difficult datasets*; and, more importantly, the lack of "generalizable" robustness, i.e., obtaining robust models against *threat models* not encountered during training. CCAT achieves comparable or better robustness against the threat model at training time, i.e., $L_\infty$-constrained adversarial examples, with better accuracy on Cifar10. However, in strong contrast to adversarial training, CCAT generalizes to stronger $L_\infty$ attacks *as well as $L_2$, $L_1$, $L_0$ attacks and distal adversarial examples by allowing detection based on confidence-thresholding. For our experiments, we further introduced a generalized robust test error adapted to our detection-setting and used a worst-case evaluation methodology, both of independent interest to the community.*

| | Corrupted MNIST: **Detection** $\tau$@99%TPR (training: $L_\infty$, $\epsilon = 0.30$) | | Corrupted CIFAR10: **Detection** $\tau$@99%TPR (training: $L_\infty$, $\epsilon = 0.03$) | |
|---|---|---|---|---|
| | Average Results over 15 Corruptions | | Average Results over 15 Corruptions | |
| | ROC AUC | Err in % | ROC AUC | Err in % |
| Normal | 0.75 | 32.8 | 0.57 | 12.3 |
| AT | 0.80 | 12.6 | 0.53 | 16.2 |
| CCAT | **0.95** | **4.0** | **0.66** | **8.5** |

Table 5: ***Results on Corrupted MNIST and Cifar10.*** *Performance on corrupted examples (spatial transformations, brightness/contrast changes, noise etc.). We report ROC AUC and confidence-thresholded Err on corrupted examples for $\tau$@99%TPR. Compared to normal training and AT, CCAT detects better corrupted examples (higher ROC AUC) and its test error Err on non-rejected examples is lower.*

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

# A    PROOF OF PROPOSITION 1

**Proposition 2.** *We consider a classification problem with two points $x = 0$ and $x = \epsilon$ in $\mathbb{R}$ with deterministic labels, that is $p(y = 2|x = 0) = 1$ and $p(y = 1|x = \epsilon) = 1$ and the problem is fully determined by the probability $p_0 = p(x = 0)$ as $p(x = \epsilon) = 1 - p_0$. The Bayes error of this classification problem is zero. The predicted probability distribution over the classes is $\tilde{p}(y|x) = \frac{e^{g_y(x)}}{e^{g_1(x)} + e^{g_2(x)}}$, where $g : \mathbb{R}^d \to \mathbb{R}^2$ and we assume that the function $\lambda : \mathbb{R}_+ \to [0,1]$ is monotonically decreasing and $\lambda(0) = 1$. The Bayes optimal classifier for the cross-entropy loss of*

- *adversarial training on $100\%$ adversarial examples, cf. Eq. (1), yields an error of $\min\{p_0, 1 - p_0\}$.*

- *adversarial training with $50\%$ adversarial and $50\%$ clean examples per batch, cf. Eq. (3), yields an error of $\min\{p_0, 1 - p_0\}$.*

- *CCAT on $50\%$ clean and $50\%$ adversarial examples, cf. Alg. 1, yields zero error if $\lambda(\epsilon) < \min\left\{\frac{p_0}{1-p_0}, \frac{1-p_0}{p_0}\right\}$.*

*Proof. First, we stress that we are dealing with three different probability distributions over the labels: the true one $p(y|x)$, the imposed one during training $\hat{p}(y|x)$ and the predicted one $\tilde{p}(y|x)$. We also note that $\hat{p}$ depends on $\lambda$ through Eq. (5), and $\lambda$ itself is a function of the norm $\|\delta\|$ through Eq. (6); here, this dependence is made explicit by writing $\hat{p}(\lambda)(y|x)$. This makes the expressions for the expected loss of CCAT slightly more complicated. We first derive the Bayes optimal classifier and its loss for CCAT.* We introduce

$$a = g_1(0) - g_2(0), \quad b = g_1(\epsilon) - g_2(\epsilon). \tag{8}$$

and express the *logarithm of the predicted probabilities (confidences)* of class 1 and 2 in terms of these quantities.

$$-\log \tilde{p}(y = 2|x = x) = -\log\left(\frac{e^{g_2(x)}}{e^{g_1(x)} + e^{g_2(x)}}\right) = \log\left(1 + e^{g_1(x) - g_2(x)}\right) = \begin{cases} \log\left(1 + e^a\right) & \text{if } x = 0 \\ \log(1 + e^b) & \text{if } x = \epsilon \end{cases}.$$

$$-\log \tilde{p}(y = 1|x = x) = -\log\left(\frac{e^{g_1(x)}}{e^{g_1(x)} + e^{g_2(x)}}\right) = \log\left(1 + e^{g_2(x) - g_1(x)}\right) = \begin{cases} \log(1 + e^{-a}) & \text{if } x = 0 \\ \log\left(1 + e^{-b}\right) & \text{if } x = \epsilon \end{cases}.$$

For our confidence-calibrated adversarial training one first has to solve

$$\delta_x^*(j) = \underset{\|\delta\|_\infty \leq \epsilon}{\mathrm{argmax}} \ \underset{k \neq j}{\max} \ \tilde{p}(y = k \,|\, x + \delta).$$

*We get with the expressions above (note that we have a binary classification problem):*

$$\delta_0^*(1) = \underset{\|\delta\|_\infty \leq \epsilon}{\mathrm{argmax}} \ \tilde{p}(y = 2|0 + \delta) = \begin{cases} 0 & \text{if } a > b \\ \epsilon & \text{else.} \end{cases}.$$

$$\delta_0^*(2) = \underset{\|\delta\|_\infty \leq \epsilon}{\mathrm{argmax}} \ \tilde{p}(y = 1|0 + \delta) = \begin{cases} \epsilon & \text{if } a > b \\ 0 & \text{else.} \end{cases}.$$

$$\delta_\epsilon^*(1) = \underset{\|\delta\|_\infty \leq \epsilon}{\mathrm{argmax}} \ \tilde{p}(y = 2|\epsilon + \delta) = \begin{cases} -\epsilon & \text{if } a > b \\ 0 & \text{else.} \end{cases}.$$

$$\delta_\epsilon^*(2) = \underset{\|\delta\|_\infty \leq \epsilon}{\mathrm{argmax}} \ \tilde{p}(y = 1|\epsilon + \delta) = \begin{cases} 0 & \text{if } a > b \\ -\epsilon & \text{else.} \end{cases}.$$

*Note that in CCAT for $50\%$ of the batch we use the standard cross-entropy loss and for the other $50\%$ we use*

$$L\left(\hat{p}_y\big(\lambda(\|\delta_x^*(y)\|)\big)(x), \tilde{p}(x)\right) = -\sum_{j=1}^{2} \hat{p}_y\big(\lambda(\|\delta_x^*(y)\|)\big)(y = j \,|\, x = x) \log\left(\tilde{p}(y = j \,|\, x = x + \delta_x^*(j))\right).$$

*The corresponding expected loss is then given by*

$$\mathbb{E}\Big[L\Big(\hat{p}_y\big(\lambda(\|\delta_x^*(y)\|)\big)(x), \tilde{p}(x)\Big)\Big] = \mathbb{E}\Big[\mathbb{E}\Big[L\big(\hat{p}_y\big(\lambda(\|\delta_x^*(y)\|)\big)(x), \tilde{p}(x)\big)|x\Big]\Big]$$

$$=p(x=0)\mathbb{E}\Big[L(\hat{p}_y(\lambda(\|\delta_0^*(y)\|))(0), \tilde{p}(0))|x=0\Big] + p(x=\epsilon)\mathbb{E}\Big[L\big(\hat{p}_Y\big(\lambda(\|\delta_\epsilon^*(y)\|)\big)(\epsilon), \tilde{p}(\epsilon)\big)|x=\epsilon\Big],$$

*where $p(x=0) = p_0$ and $p(x=\epsilon) = 1-p(x=0) = 1-p_0$. With the true conditional probabilities $p(y=s\,|\,x=x)$ we get*

$$\mathbb{E}[L(\hat{p}_y(\lambda(\delta))(x), \tilde{p}(x))|x=x] = \sum_{s=1}^{2} p(y=s\,|\,x=x)L(\hat{p}_s(\lambda(\|\delta_x^*(s)\|))(x), \tilde{p}(x))$$

$$= -\sum_{s=1}^{2} p(y=s\,|\,x=x)\sum_{j=1}^{2}\hat{p}_s(\|\lambda(\delta_x^*(s)\|))(y=j\,|\,x=x)\log\big(\tilde{p}(y=j\,|\,x=x+\delta_x^*(s))\big)$$

*For our problem it holds with $p(y=2\,|\,x=0) = p(y=1\,|\,x=\epsilon) = 1$ (by assumption). Thus,*

$$\mathbb{E}[L(\hat{p}_y(\lambda(\delta))(x), \tilde{p}(x))|x=0] = -\sum_{j=1}^{2}\hat{p}_2(\lambda(\|\delta_0^*(2)\|))(y=j\,|\,x=0)\log\big(\tilde{p}(y=j\,|\,x=x+\delta_0^*(2))\big)$$

$$\mathbb{E}[L(\hat{p}_y(\lambda(\delta))(x), \tilde{p}(x))|x=\epsilon] = -\sum_{j=1}^{2}\hat{p}_1(\lambda(\|\delta_\epsilon^*(1)\|))(y=j\,|\,x=\epsilon)\log\big(\tilde{p}(y=j\,|\,x=x+\delta_\epsilon^*(1))\big)$$

*As $\|\delta_x^*(y)\|$ is either $0$ or $\|\epsilon\|$ and $\lambda(0) = 1$ we use in the following for simplicity the notation $\lambda = \lambda(\|\epsilon\|)$. Moreover, note that*

$$\hat{p}_y(\lambda)(y=j|x=x) = \begin{cases} \lambda + \frac{(1-\lambda)}{K} & \textit{if } y=j, \\ \frac{(1-\lambda)}{K} & \textit{else} \end{cases},$$

*where $K$ is the number of classes. Thus, $K=2$ in our example and we note that $\lambda + \frac{(1-\lambda)}{2} = \frac{1+\lambda}{2}$.*

With this we can write the total loss (remember that we have half normal cross-entropy loss and half the loss for the adversarial part with the modified "labels") as

$$L(a,b) = p_0\Big[\log(1+e^a)\mathbb{1}_{a\geq b} + \mathbb{1}_{a<b}\Big(\frac{(1+\lambda)}{2}\log(1+e^b) + \frac{(1-\lambda)}{2}\log(1+e^{-b})\Big)\Big]$$

$$+ (1-p_0)\Big[\log(1+e^{-b})\mathbb{1}_{a\geq b} + \mathbb{1}_{a<b}\Big(\frac{(1+\lambda)}{2}\log(1+e^{-a}) + \frac{(1-\lambda)}{2}\log(1+e^a)\Big)\Big]$$

$$+ \log(1+e^a)p_0 + \log(1+e^{-b})(1-p_0),$$

where we have omitted a global factor $\frac{1}{2}$ for better readability *(the last row is the cross-entropy loss)*. We distinguish two sets in the optimization. First we consider the case $a \geq b$. Then it is easy to see that in order to minimize the loss we have $a = b$.

$$\partial_a L = 2\frac{e^a}{1+e^a}p_0 - \frac{e^{-a}}{1+e^{-a}}(1-p_0)$$

This yields $e^a = \frac{1-p_0}{p_0}$ or $a = \log\left(\frac{1-p_0}{p_0}\right)$ and the minimum for $a \geq b$ is attained on the boundary of the domain of $a \leq b$. The other case is $a \leq b$. We get

$$\partial_a L = \Big[\frac{(1+\lambda)}{2}\frac{-e^{-a}}{1+e^{-a}} + \frac{(1-\lambda)}{2}\frac{e^a}{1+e^a}\Big](1-p_0) + p_0\frac{e^a}{1+e^a}$$

$$\partial_b L = \Big[\frac{(1+\lambda)}{2}\frac{e^b}{1+e^b} + \frac{(1-\lambda)}{2}\frac{-e^b}{1+e^{-b}}\Big]p_0 + (1-p_0)\frac{-e^{-b}}{1+e^{-b}}$$

This yields the solution

$$a^* = \log\left(\frac{\frac{1+\lambda}{2}(1-p_0)}{p_0 + \frac{1-\lambda}{2}(1-p_0)}\right), \qquad b^* = \log\left(\frac{\frac{1-\lambda}{2}p_0 + (1-p_0)}{\frac{1+\lambda}{2}p_0}\right)$$

It is straightforward to check that $a^* < b^*$ for all $0 < p_0 < 1$, *indeed we have*

$$\frac{\frac{1+\lambda}{2}(1-p_0)}{p_0 + \frac{1-\lambda}{2}(1-p_0)} = \frac{\frac{1+\lambda}{2}(1-p_0)}{p_0\frac{1+\lambda}{2} + \frac{1-\lambda}{2}} = \frac{1 - p_0 - \frac{(1-\lambda)}{2}(1-p_0)}{p_0\frac{1+\lambda}{2} + \frac{1-\lambda}{2}} < \frac{\frac{1-\lambda}{2}p_0 + (1-p_0)}{\frac{1+\lambda}{2}p_0}$$

*if $0 < p_0 < 1$ and note that $\lambda < 1$ by assumption.* We have $a^* < 0$ *and thus $g_2(0) > g_1(0)$ (Bayes optimal decision for $x = 0$)* if

$$1 > \frac{1-p_0}{p_0}\lambda,$$

and $b^* > 0$ *and thus $g_1(\epsilon) > g_2(\epsilon)$ (Bayes optimal decision for $x = \epsilon$)* if

$$1 > \frac{p_0}{1-p_0}\lambda.$$

Thus we recover the Bayes classifier if

$$\lambda < \min\left\{\frac{1-p_0}{p_0}, \frac{p_0}{1-p_0}\right\}.$$

Now we consider the approach by Madry et al. (2018) with $100\%$ adversarial training. The expected loss can be written as

$$\mathbb{E}\left[\max_{\|\delta\|\leq\infty} L(y, f(x+\delta))\right] = \mathbb{E}\left[\mathbb{E}\left[\max_{\|\delta\|\leq\infty} L(y, f(x+\delta)|x)\right]\right]$$

$$= p(x=0)p(y=2|x=x)\max\left\{-\log\left(\tilde{p}(y=2|x=0)\right), -\log\left(\tilde{p}(y=2|x=\epsilon)\right)\right\}$$

$$+ (1-p(x=0))p(y=1|x=x)\max\left\{-\log\left(\tilde{p}(y=1|x=0)\right), -\log\left(\tilde{p}(y=1|x=\epsilon)\right)\right\}$$

$$= p(x=0)\max\left\{-\log\left(\tilde{p}(y=2|x=0)\right), -\log\left(\tilde{p}(y=2|x=\epsilon)\right)\right\}$$

$$+ (1-p(x=0))\max\left\{-\log\left(\tilde{p}(y=1|x=0)\right), -\log\left(\tilde{p}(y=1|x=\epsilon)\right)\right\}$$

*This yields in terms of the parameters $a, b$ the expected loss:*

$$L(a,b) = \max\left\{\log(1+e^a), \log(1+e^b)\right\}p_0 + \max\left\{\log(1+e^{-a}), \log(1+e^{-b})\right\}(1-p_0)$$

The expected loss is minimized if $a = b$ as then both maxima are minimal. This results in the expected loss

$$L(a) = \log(1+e^a)p_0 + \log(1+e^{-a})(1-p_0).$$

The critical point is attained at $a^* = b^* = \log\left(\frac{1-p_0}{p_0}\right)$. This is never Bayes optimal for all $0 < p_0 < 1$. *Note that $b^* = a^* > 0$ if $p_0 < 1 - p_0$ and thus the error is given by $p_0$, similar $a^* = b^* < 0$ if $1 - p_0 < p_0$ and thus the error is given by $1 - p_0$. We get an error of $\min\{p_0, 1-p_0\}$ of the Bayes optimal solution of $100\%$ adversarial training.*

Next we consider $50\%$ adversarial plus $50\%$ clean training. The expected loss

$$\mathbb{E}\left[\max_{\|\delta\|\leq\infty} L(y, f(x+\delta))\right] + \mathbb{E}\left[L(y, f(x+\delta))\right],$$

can be written as

$$L(a,b) = \max\left\{\log(1+e^a), \log(1+e^b)\right\}p_0$$

$$+ \max\left\{\log(1+e^{-a}), \log(1+e^{-b})\right\}(1-p_0)$$

$$+ \log(1+e^a)p_0 + \log(1+e^{-b})(1-p_0)$$

We make a case distinction. If $a \geq b$, then the loss reduces to

$$L(a,b) = \log(1+e^a)p_0 + \log(1+e^{-b})(1-p_0)$$

$$+ \log(1+e^a)p_0 + \log(1+e^{-b})(1-p_0)$$

$$\geq L(a,a)$$

$$= 2\log(1+e^a)p_0 + 2\log(1+e^{-a})(1-p_0)$$

| **MNIST:** Attack Ablation with RErr for $\tau$@99%TPR $(L_\infty$ attack with $\epsilon = 0.3)$ | | | | | | | | | |
|---|---|---|---|---|---|---|---|---|---|
| Optimization | momentum+backtrack | | | | | momentum | | – | |
| Initialization | zero | | | | rand | zero | | zero | |
| Iterations $T$ | 40 | 200 | 2000 | 4000 | 4000 | 60 | 300 | 60 | 300 |
| AT | 0.4 | 0.4 | 0.4 | 0.3 | 0.6 | 0.4 | 0.6 | 0.4 | 0.4 |
| AT Conf | 0.8 | 0.8 | 0.8 | 0.8 | 1.1 | 1.0 | 1.1 | 1.0 | 1.0 |
| CCAT | 0.6 | 3.3 | 4.9 | 6.8 | 5.5 | 0.9 | 3.8 | 0.0 | 0.1 |

| **SVHN:** Attack Ablation with RErr for $\tau$@99%TPR $(L_\infty$ attack with $\epsilon = 0.03)$ | | | | | | | | | |
|---|---|---|---|---|---|---|---|---|---|
| Optimization | momentum+backtrack | | | | | momentum | | – | |
| Initialization | zero | | | | rand | zero | | zero | |
| Iterations $T$ | 40 | 200 | 2000 | 4000 | 4000 | 60 | 300 | 60 | 300 |
| AT | 38.4 | 46.2 | 49.9 | 50.1 | 51.8 | 37.7 | 38.1 | 29.9 | 30.8 |
| AT Conf | 27.4 | 40.5 | 46.9 | 47.3 | 48.1 | 27.1 | 28.5 | 21.1 | 23.8 |
| CCAT | 4.0 | 5.0 | 22.8 | 23.3 | 5.2 | 2.6 | 2.6 | 2.6 | 2.6 |

| **Cifar10:** Attack Ablation with RErr for $\tau$@99%TPR $(L_\infty$ attack with $\epsilon = 0.03)$ | | | | | | | | | |
|---|---|---|---|---|---|---|---|---|---|
| Optimization | momentum+backtrack | | | | | momentum | | – | |
| Initialization | zero | | | | rand | zero | | zero | |
| Iterations $T$ | 40 | 200 | 2000 | 4000 | 4000 | 60 | 300 | 60 | 300 |
| AT | 60.9 | 60.8 | 60.8 | 60.8 | 60.9 | 60.9 | 60.9 | 57.4 | 57.6 |
| AT Conf | 60.4 | 60.6 | 60.5 | 60.5 | 60.9 | 60.4 | 60.6 | 56.2 | 56.6 |
| CCAT | 14.8 | 16.2 | 40.2 | 41.3 | 34.9 | 7.2 | 7.2 | 7.2 | 7.2 |

Table 6: **Detailed attack ablation studies on MNIST, SVHN and Cifar10.** Complementary to Tab. 1, we compare our $L_\infty$ PGD-Conf, *as introduced in Sec. 4.1*, attack with $T$ iterations and different combinations of momentum, backtracking and initialization on all three datasets. We consider AT, AT trained with PGD-Conf, and CCAT; we report RErr for confidence threshold $\tau$@99%TPR. As backtracking requires an additional forward pass per iteration, we use $T = 60$ and $T = 300$ for attacks without backtracking to be comparable to attacks with $T = 40$ and $T = 200$ with backtracking.

Solving for the critical point yields $a^* = \log\left(\frac{1-p_0}{p_0}\right) = b^*$. Next we consider the set $a \leq b$. This yields the loss

$$L(a,b) = \log(1 + e^b)p_0 + \log(1 + e^{-a})(1 - p_0)$$
$$+ \log(1 + e^a)p_0 + \log(1 + e^{-b})(1 - p_0)$$

Solving for the critical point yields $a^* = \log\left(\frac{1-p_0}{p_0}\right) = b^*$ which fulfills $a \leq b$. Actually, it coincides with the solution found already for $100\%$ adversarial training. One does not recover the Bayes classifier for any $0 < p_0 < 1$.

Moreover, we note that

$$a^* = b^* = \begin{cases} > 0 & \text{if } p_0 < \frac{1}{2}, \\ < 0 & \text{if } p_0 > \frac{1}{2}. \end{cases}$$

Thus, we classify $x = 0$ correctly, if $p_0 > \frac{1}{2}$ and $x = \epsilon$ correctly if $p_0 < \frac{1}{2}$. Thus the error is given by $\min\{p_0, 1 - p_0\}$. $\qquad\square$

# B  EXPERIMENTS

We give additional details on our experimental setup, specifically regarding attacks, training and the used evaluation metrics. Afterwards, we include additional experimental results, including ablation studies, results for $98\%$ true positive rate (TPR), results per attack, *results per corruption on MNIST-C (Mu & Gilmer, 2019) and Cifar10-C (Hendrycks & Dietterich, 2019) and a comparison with (Pang et al., 2018)*.

---

**Algorithm 2** Pseudo-code for the used projected gradient descent (PGD) procedure to maximize Eq. (9) or Eq. (10) subject to the constraints $\tilde{x}_i = x_i + \delta_i \in [0, 1]$ and $\|\delta\|_\infty \leq \epsilon$; in practice, the procedure is applied on batches of inputs. The algorithm is also easily adapted to work with a $L_2$-norm; only the projections on line 6 and 24 needs to be adapted.

---

    **input:** example $x$ with label $y$
    **input:** number of iterations $T$
    **input:** learning rate $\gamma$, momentum $\beta$, learning rate factor $\alpha$
    **input:** initial $\delta^{(0)}$, e.g., Eq. (11) or $\delta^{(0)} = 0$
1:  $v := 0$ {best objective achieved}
2:  $\tilde{x} := x + \delta^{(0)}$ {best adversarial example}
3:  $g^{(-1)} := 0$ {accumulated gradients}
4:  **for** $t = 0, \ldots, T$ **do**
5:     {projection onto $L_\infty$ $\epsilon$-ball and on $[0, 1]$:}
6:     clip $\delta_i^{(t)}$ to $[-\epsilon, \epsilon]$
7:     clip $x_i + \delta_i^{(t)}$ to $[0, 1]$
8:     {forward and backward pass to get objective and gradient:}
9:     $v^{(t)} := \mathcal{F}(x + \delta^{(t)}, y)$
10:    $g^{(t)} := \text{sign}\left(\nabla_{\delta^{(t)}} \mathcal{F}(x + \delta^{(t)}, y)\right)$
11:    {keep track of adversarial example resulting in best objective:}
12:    **if** $v^{(t)} > v$ **then**
13:       $v := v^{(t)}$
14:       $\tilde{x} := x + \delta^{(t)}$
15:    **end if**
16:    {iteration $T$ is only meant to check whether last update improved objective:}
17:    **if** $t = T$ **then**
18:       **break**
19:    **end if**
20:    {integrate momentum term:}
21:    $g^{(t)} := \beta g^{(t-1)} + (1 - \beta)g^{(t)}$
22:    {"try" the update step and see if objective increases:}
23:    $\hat{\delta}^{(t)} := \delta^{(t)} + \gamma g^{(t)}$
24:    clip $\hat{\delta}_i^{(t)}$ to $[-\epsilon, \epsilon]$
25:    clip $x_i + \hat{\delta}_i^{(t)}$ to $[0, 1]$
26:    $\hat{v}^{(t)} := \mathcal{F}(x + \hat{\delta}^{(t)}, y)$
27:    {only keep the update if the objective increased; otherwise decrease learning rate:}
28:    **if** $\hat{v}^{(t)} \geq v^{(t)}$ **then**
29:       $\delta^{(t+1)} := \hat{\delta}^{(t)}$
30:    **else**
31:       $\gamma := \gamma/\alpha$
32:    **end if**
33: **end for**
34: **return** $\tilde{x}, \tilde{v}$

---

### B.1 ATTACKS

Complementary to the description of the projected gradient descent (PGD) attack by Madry et al. (2018) and our adapted attack, we provide a detailed algorithm in Alg. 2. We note that the objective maximized in (Madry et al., 2018) is

$$\mathcal{F}(x + \delta, y) = \mathcal{L}(f(x + \delta; w), y) \tag{9}$$

where $\mathcal{L}$ denotes the cross-entropy loss, $f(\cdot; w)$ denotes the model and $(x, y)$ is an input-label pair from the test set. Our adapted attack, in contrast, maximizes

$$\mathcal{F}(x + \delta, y) = \max_{k \neq y} f_k(x + \delta; w). \tag{10}$$

*Note that the maximum over labels, i.e., $\max_{k \neq y}$ is explicitly computed during optimization; this means that in contrast to (Goodfellow et al., 2019), we do not run $K$ targeted attack and subse-*

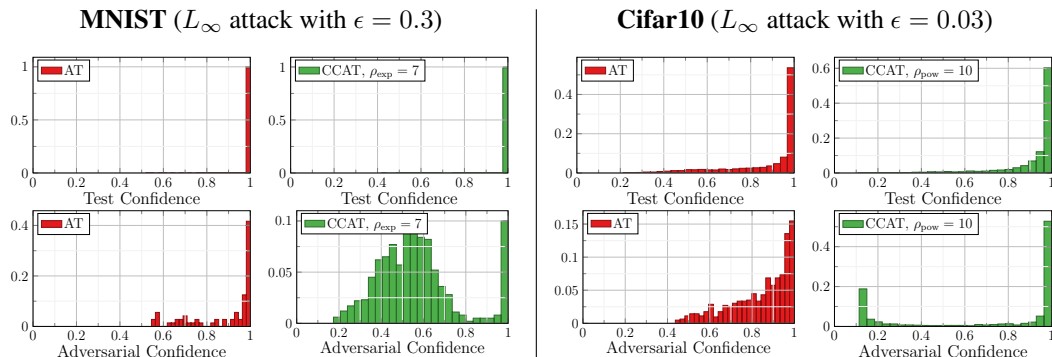

Figure 5: **Confidence histograms on MNIST and Cifar10.** As in Fig. 3, we show histograms of confidences on correctly classified test examples (top) and on successful adversarial examples (bottom) for both AT and CCAT. Note that on AT, the number of successful adversarial examples is usually lower than on CCAT, i.e., reflects the RErr for $\tau = 0$ in Tab. 3; for CCAT in contrast, nearly all adversarial examples are successful, while only a part has high confidence. Histograms obtained for the worst-case adversarial examples across all tested $L_\infty$ attacks.

*quently take the maximum-confidence one, where $K$ is the number of classes.* We denote these two variants as PGD-CE and PGD-Conf, respectively. Deviating from (Madry et al., 2018), we initialize $\delta$ uniformly over directions and norm (instead of uniform initialization over the volume of the $\epsilon$-ball):

$$\delta = u\epsilon \frac{\delta'}{\|\delta'\|_\infty}, \quad \delta' \sim \mathcal{N}(0, I), u \sim U(0, 1) \tag{11}$$

where $\delta'$ is sampled from a standard Gaussian and $u \in [0, 1]$ from a uniform distribution. We also consider zero initialization, i.e., $\delta = 0$. For random initialization we always consider multiple attempts, 10 for PGD-Conf and 50 for PGD-CE, with zero initialization, we use only 1 attempt.

*Both PGD-CE and PGD-Conf can also be applied using the $L_2$, $L_1$ and $L_0$ norms following the description above. Then, gradient normalization, i.e., the signum in Line 10 in Alg. 2 for the $L_\infty$ norm, the projection, and the the initialization need to be adapted. For the $L_2$ norm, the gradient is normalized by dividing by the $L_2$ norm; for the $L_1$ norm only the $1\%$ largest values (in absolute terms) of the gradient are kept and normalized by their $L_1$ norm; and for the $L_0$ norm, the gradient is normalized by the $L_1$ norm. Projection is easily implemented for the $L_2$ norm, while we follow the algorithm of Duchi et al. (2008) for the $L_1$ project; for the $L_0$ projection, only the $\epsilon$-largest values are kept. Similarly, initialization for $L_2$ and $L_1$ are simple by randomly choosing a direction (as in Eq. (11)) and then normalizing by their norm. For $L_0$, we randomly choose pixels with probability $2/3\epsilon/(HWD)$ and set them to a uniformly random values $u \in [0, 1]$, where $H \times W \times D$ is the image size. In experiments, we found that tuning the learning rate for PGD with $L_1$ and $L_0$ constraints (independent of the objective, i.e., Eq. (9) or Eq. (10)) is much more difficult. Additionally, PGD using the $L_0$ norm seems to get easily stuck in sub-optimal local optima.*

Alg. 2 also gives more details on the employed momentum and backtracking scheme. These two "tricks" add two additional hyper-parameters to the number of iterations $T$ and the learning rate $\gamma$, namely the momentum parameter $\beta$ and the learning rate factor $\alpha$. After each iteration, the computed update, already including the momentum term, is only applied if this improves the objective. This is checked through an additional forward pass. If not, the learning rate is divided by $\alpha$, and the update is rejected. Alg. 2 includes this scheme as an algorithm for an individual test example $x$ with label $y$ for brevity; however, extending it to work on batches, which is used in our paper, is straight-forward. In practice, for PGD-CE, with $T = 200$ iterations, we use $\beta = 0.9$ and $\alpha = 1.25$; for PGD-Conf, with $T = 2000$ iterations, we use $\beta = 0.9$ and $\alpha = 1.1$.

We also give more details on the used black-box attacks. For random sampling, we apply Eq. (11) $T = 5000$ times in order to maximize Eq. (10). We also implemented the black-box attack of Ilyas et al. (2018a) using a population of 50 and variance of 0.1 for estimating the gradient in Line 10 of Alg. 2; a detailed algorithm is provided in (Ilyas et al., 2018a). We use a learning rate of 0.001 (note that the gradient is signed, as in (Madry et al., 2018)) and also integrated a momentum with $\beta = 0.9$

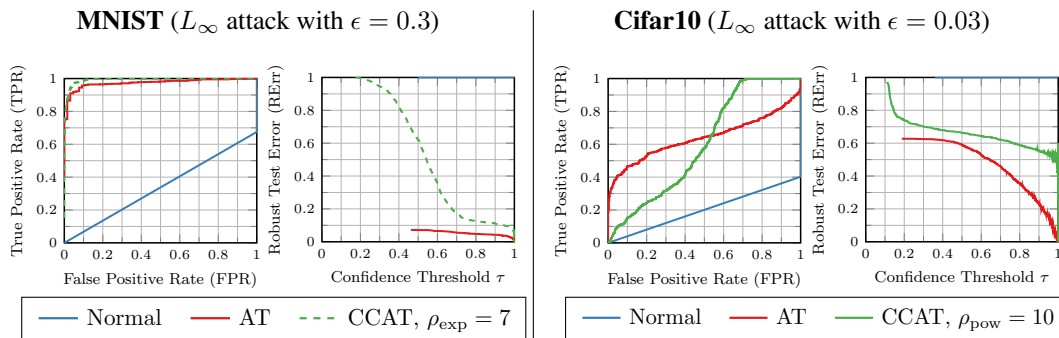

Figure 6: **ROC and RErr curves on MNIST and Cifar10.** ROC curves, i.e. FPR plotted against TPR for all possible confidence thresholds $\tau$, and RErr curves, i.e., RErr over confidence threshold $\tau$ for AT and CCAT, including different $\rho$ parameters. Worst-case adversarial examples across all $L_\infty$ attacks were tested.

and backtracking as described in Alg. 2 with $\alpha = 1.1$ and $T = 2000$ iterations. We use zero and random initialization; in the latter case we allow 10 random retries. For the simple black-box attack we follow the algorithmic description in (Narodytska & Kasiviswanathan, 2017) considering only axis-aligned perturbations of size $\epsilon$ per pixel. We run the attack for $T = 2000$ iterations and allow 10 random retries. Finally, we use a variant of the cube attack proposed in (Andriushchenko, 2019). We run the attack for $T = 5000$ iterations with a probability of change of 0.05. We emphasize that, except for (Ilyas et al., 2018a), these attacks are not gradient-based and do not approximate the gradient.

## B.2 TRAINING

As described in Sec. 4, we follow the ResNet-20 architecture by He et al. (2016) implemented in PyTorch (Paszke et al., 2017). For training we use a batch size of 100 and train for 100 and 200 epochs on MNIST and SVHN/Cifar10, respectively: this holds for normal training, adversarial training (AT) and confidence-calibrated adversarial training (CCAT). For the latter two, we use PGD-CE and PGD Conf, respectively, for $T = 40$ iterations, momentum and backtracking ($\beta = 0.9$, $\alpha = 1.5$). For PGD-CE we use a learning rate of 0.05, 0.01 and 0.005 on MNIST, SVHN and Cifar10. For PGD-Conf we use a learning rate of 0.05. For training, we use standard stochastic gradient descent, starting with a learning rate of 0.1 on MNIST/SVHN and 0.075 on Cifar10. The learning rate is multiplied by 0.95 after each epoch. We do not use weight decay; but the network includes batch normalization (Ioffe & Szegedy, 2015). On SVHN and Cifar10, we use random cropping, random flipping (only Cifar10) and contrast augmentation during training. We always train on 50% clean and 50% adversarial examples per batch, i.e., each batch contains both clean and adversarial examples which is important when using batch normalization.

## B.3 EVALUATION METRICS

For reproducibility and complementing the discussion in the main paper, we describe the used evaluation metrics and evaluation procedure in more detail. Adversarial examples are computed on the first 1000 examples of the test set; the used confidence threshold is computed on the last 1000 examples of the test set; test errors are computed on all test examples minus the last 1000. As we consider multiple attacks, and some attacks allow multiple random attempts, we always consider the worst case adversarial example per test example and across all attacks/attempts; the worst-case is selected based on confidence.

**ROC AUC:** To compute ROC curves, and the area under the curve, i.e., ROC AUC, we define negatives as *successful* adversarial examples (on correctly classified test examples) and positives as the corresponding *correctly classified* test examples. The ROC AUC as well as the curve itself can easily be calculated using (Pedregosa et al., 2011). Practically, the generated curve could be used to directly estimate a threshold corresponding to a pre-determined true positive rate (TPR). However, this requires interpolation; after trying several constant interpolation schemes, we concluded that the

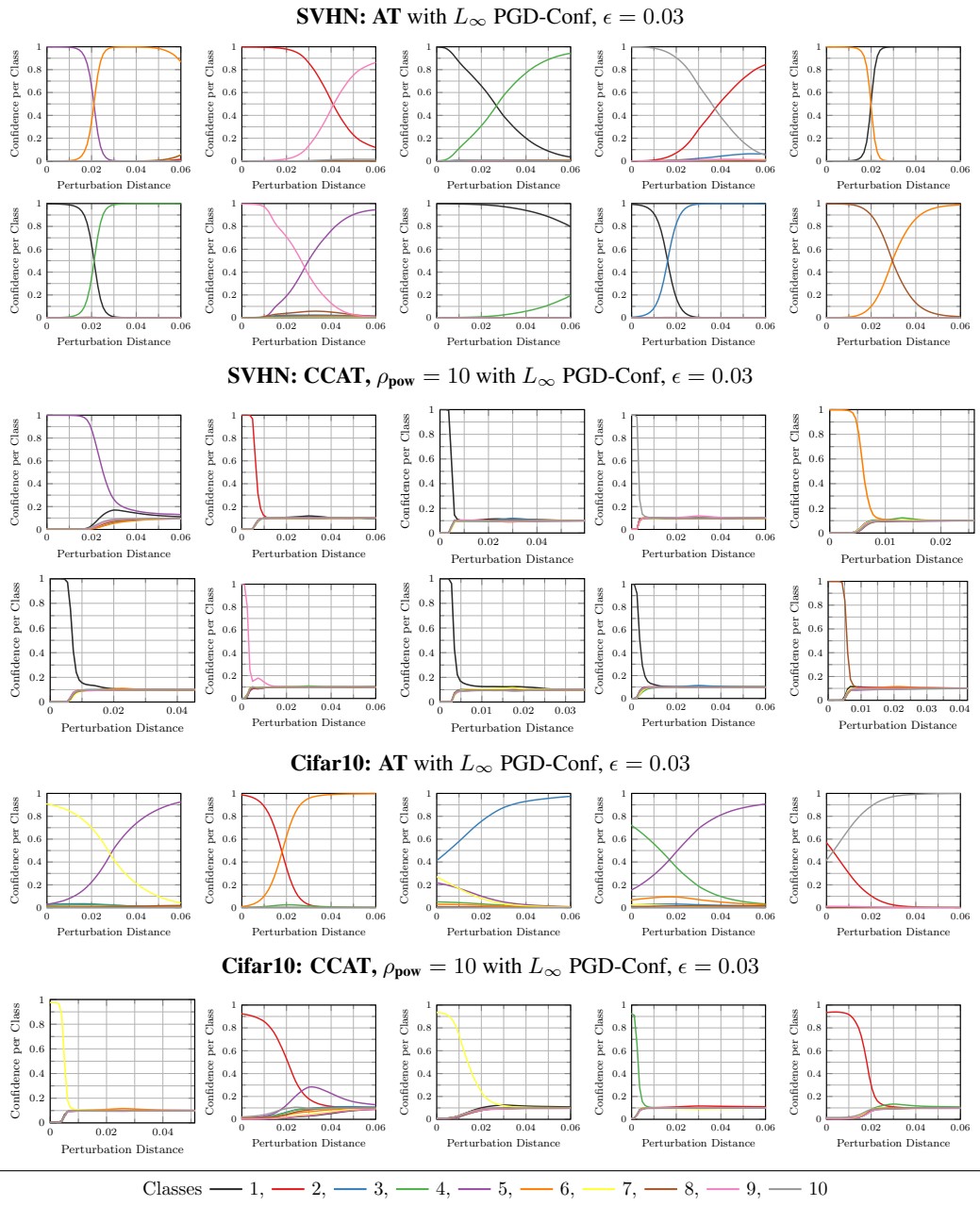

Figure 7: **Effect of confidence calibration on SVHN and Cifar10.** Confidences for classes along adversarial directions for AT and CCAT, $\rho_{\text{pow}} = 10$. Adversarial examples were computed using PGD-Conf with $T = 2000$ iterations and $\gamma = 0.001$ and zero initialization. For both AT and CCAT, we show the first ten examples of the test set on SVHN, and the first five examples of the test set on Cifar10.

results are distorted significantly, especially for TPRs close to $100\%$. Thus, we followed a simpler scheme as mentioned above: on a held out validation set of size 1000 (the last 1000 samples of the test set), we sorted the corresponding confidences, and picked the confidence threshold in order to obtain the desired TPR, e.g., $99\%$, on this set exactly.

**MNIST** (PGD-Conf, $L_\infty$ with $\epsilon = 0.3$)

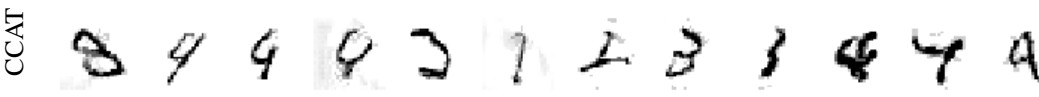

**SVHN** (PGD-Conf, $L_\infty$ with $\epsilon = 0.03$)

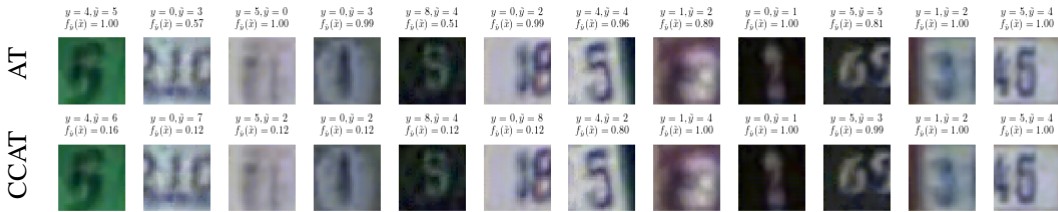

**Cifar10** (PGD-Conf, $L_\infty$ with $\epsilon = 0.03$)

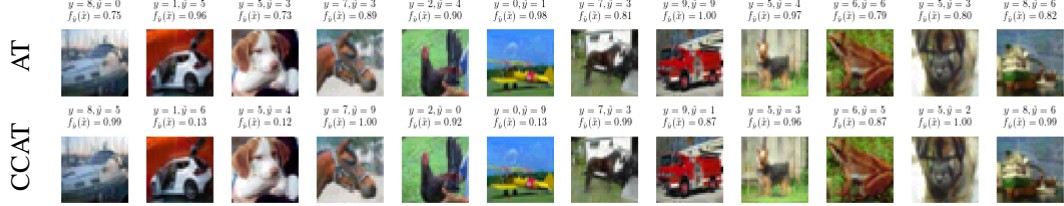

Figure 8: *Adversarial examples against AT and CCAT on MNIST, SVHN, and Cifar10. $L_\infty$-PGD-Conf adversarial examples for MNIST, $\epsilon = 0.3$, and SVHN/Cifar10, $\epsilon = 0.03$, for AT and CCAT. In all cases, twelve test examples have been attacked. On the first six examples, AT has difficulties, on the last six examples, CCAT has difficulties. We also report the true label $y$, the target label $\tilde{y}$ of the adversarial examples and the corresponding confidence $f_{\tilde{y}}(\tilde{x})$.*

**Robust Test Error:** For clarity, we repeat our confidence-integrated definition of the robust test error:

$$\text{RErr}(\tau) = \frac{\sum_{n=1}^{N} \mathbb{1}_{f(x_n) \neq y_n} \mathbb{1}_{c(x_n) \geq \tau} + \sum_{n=1}^{N} \mathbb{1}_{f(x_n) = y_n} \mathbb{1}_{f(\tilde{x}_n) \neq y_n} \mathbb{1}_{c(\tilde{x}_n) \geq \tau}}{\sum_{n=1}^{N} \mathbb{1}_{c(x_n) \geq \tau} + \sum_{n=1}^{N} \mathbb{1}_{c(x_n) < \tau} \mathbb{1}_{c(\tilde{x}_n) \geq \tau} \mathbb{1}_{f(x_n) = y_n} \mathbb{1}_{f(\tilde{x}_n) \neq y_n}}, \quad (12)$$

As described in the main paper, Eq. (12) quantifies the performance of a classifier with reject-option at a specific confidence threshold $\tau$ on both clean and adversarial examples. The regular robust test error, corresponding to Eq. (12) at $\tau = 0$, can also be written as

$$\text{RErr}(0) = \frac{\sum_{n=1}^{N} \mathbb{1}_{f(x_n) \neq y_n} + \sum_{n=1}^{N} \mathbb{1}_{f(x_n) = y_n} \mathbb{1}_{f(\tilde{x}_n) \neq y_n}}{\sum_{n=1}^{N} 1}. \quad (13)$$

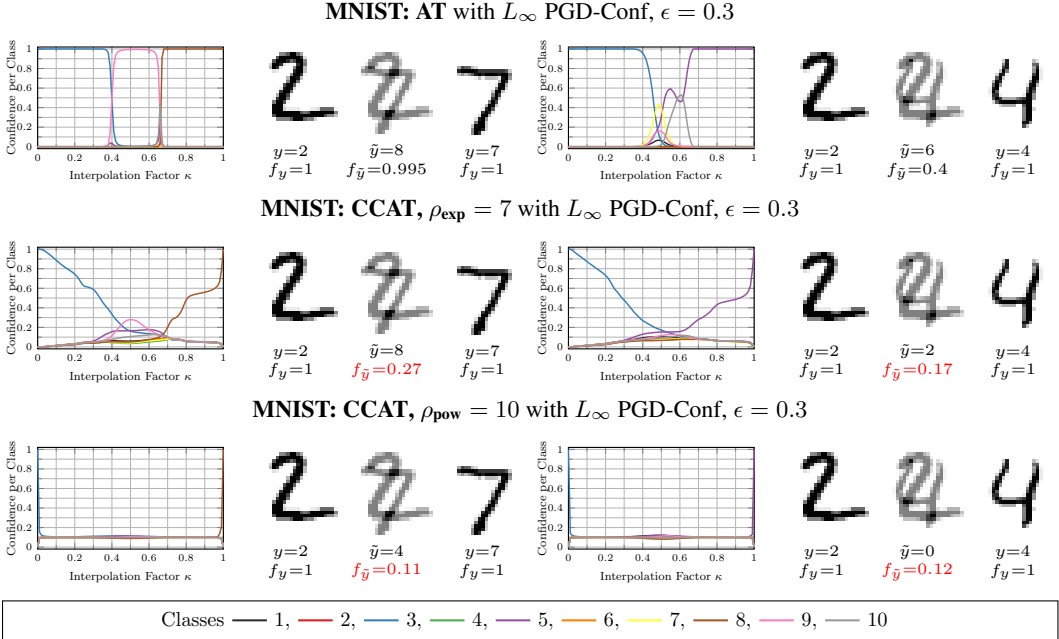

Figure 9: *Confidence calibration between test examples on MNIST. We plot the confidence for all classes when interpolating linearly between test examples: $(1 - \kappa)x_1 + \kappa x_2$ for two test examples $x_1$ and $x_2$; $x_1$ is fixed and we show two examples corresponding to different $x_2$. Additionally, we show the corresponding images for $\kappa = 0$, i.e., $x_1$, $\kappa = 0.5$, i.e., the mean image, and $\kappa = 1$, i.e., $x_2$, with the corresponding labels and confidences. As can be seen, CCAT with power transition, cf. Eq. (6), is able to perfectly predict a uniform distribution between test examples, while the exponential transition enforces a stronger bias towards the true labels.*

The robust test error is easy to handle as it quantifies overall performance, including generalization on clean examples (first term in Eq. (13)) and robustness on adversarial examples corresponding to correctly classified clean examples (second term in Eq. (13)). Additionally, the robust test error lies in $[0, 1]$. Generalizing Eq. (13) to $\tau > 0$ is non-trivial due to the following considerations: First, when integrating a confidence threshold, the reference set (i.e., the denominator) needs to be adapted. Otherwise, the metric will not reflect the actual performance after thresholding, i.e., rejecting examples – specifically, the values are not comparable across different confidence thresholds $\tau$. Then, Eq. (13) might be adapted as follows:

$$\text{RErr}(\tau) = \frac{\sum_{n=1}^{N} \mathbb{1}_{f(x_n) \neq y_n} \mathbb{1}_{c(x_n) \geq \tau} + \sum_{n=1}^{N} \mathbb{1}_{f(x_n) = y_n} \mathbb{1}_{f(\tilde{x}_n) \neq y_n} \mathbb{1}_{c(\tilde{x}_n) \geq \tau}}{\sum_{n=1}^{N} \mathbb{1}_{c(x_n) \geq \tau}}. \quad (14)$$

Note that in the nominator, we only consider clean and adversarial examples with confidence above the threshold $\tau$, i.e., $c(x_n) \geq \tau$ and $c(\tilde{x}_n) \geq \tau$. Similarly, the denominator has been adapted accordingly, as after rejection, the reference set changes, too. However, this formulation has a problem when adversarial examples obtain higher confidence than the original clean examples. Thus, second, we need to account for the case where $c(\tilde{x}_n) > \tau \geq c(x_n)$, i.e., an adversarial example obtains a higher confidence than the corresponding clean example. Then, the nominator may exceed the denominator, resulting in a value larger than one. To mitigate this problem, we need to include exactly this case in the denominator. Formalizing this case, we see that it corresponds exactly to the second term in the denominator or Eq. (12):

$$\sum_{n=1}^{N} \mathbb{1}_{c(x_n) < \tau} \mathbb{1}_{c(\tilde{x}_n) \geq \tau} \mathbb{1}_{f(x_n) = y_n} \mathbb{1}_{f(\tilde{x}_n) \neq y_n} \quad (15)$$

Overall, with Eq. (12) we obtain a metric that lies within $[0, 1]$, is comparable across thresholds $\tau$ and, for $\tau = 0$, reduces to the regular robust test error as used in related work (Madry et al., 2018).

| | $L_\infty, \epsilon = 0.30$ | | $L_\infty, \epsilon = 0.40$ | | $L_2, \epsilon = 3$ | | $L_1, \epsilon = 10.00$ | | $L_0, \epsilon = 15$ | |
|---|---|---|---|---|---|---|---|---|---|---|
| | **MNIST:** Main Results for **Detection** $\tau@98\%$TPR ($L_\infty, \epsilon = 0.30$ during training | | | | | | | | | |
| | (seen) | | (unseen) | | (unseen) | | (unseen) | | (unseen) | |
| | ROC AUC | RErr in % | ROC AUC | RErr in % | ROC AUC | RErr in % | ROC AUC | RErr in % | ROC AUC | RErr in % |
| AT | 0.97 | 0.5 | 0.20 | 100.0 | 0.73 | 67.6 | 0.93 | 3.7 | 0.99 | 0.8 |
| CCAT, $\rho_{exp}=7$ | 0.99 | 5.6 | 0.94 | 29.2 | 1.00 | 0.6 | 0.96 | 10.9 | 0.99 | 5.1 |
| CCAT, $\rho_{pow}=10$ | 0.96 | 18.1 | 0.94 | 21.2 | 1.00 | 0.0 | 1.00 | 1.1 | 0.99 | 3.5 |

| | $L_\infty, \epsilon = 0.03$ | | $L_\infty, \epsilon = 0.06$ | | $L_2, \epsilon = 1$ | | $L_1, \epsilon = 7.85$ | | $L_0, \epsilon = 10$ | |
|---|---|---|---|---|---|---|---|---|---|---|
| | **SVHN:** Main Results for **Detection** $\tau@98\%$TPR ($L_\infty, \epsilon = 0.03$ during training) | | | | | | | | | |
| | (seen) | | (unseen) | | (unseen) | | (unseen) | | (unseen) | |
| | ROC AUC | RErr in % | ROC AUC | RErr in % | ROC AUC | RErr in % | ROC AUC | RErr in % | ROC AUC | RErr in % |
| AT | 0.55 | 52.9 | 0.32 | 86.9 | 0.26 | 91.3 | 0.34 | 90.9 | 0.90 | 56.8 |
| CCAT | 0.70 | 36.5 | 0.70 | 34.8 | 0.91 | 15.6 | 0.89 | 17.6 | 1.00 | 2.1 |

| | $L_\infty, \epsilon = 0.03$ | | $L_\infty, \epsilon = 0.06$ | | $L_2, \epsilon = 1$ | | $L_1, \epsilon = 7.85$ | | $L_0, \epsilon = 10$ | |
|---|---|---|---|---|---|---|---|---|---|---|
| | **CIFAR10:** Main Results for **Detection** $\tau@98\%$TPR ($L_\infty, \epsilon = 0.03$ during training) | | | | | | | | | |
| | (seen) | | (unseen) | | (unseen) | | (unseen) | | (unseen) | |
| | ROC AUC | RErr in % | ROC AUC | RErr in % | ROC AUC | RErr in % | ROC AUC | RErr in % | ROC AUC | RErr in % |
| AT | 0.64 | 61.9 | 0.35 | 93.6 | 0.59 | 73.6 | 0.61 | 67.6 | 0.80 | 71.8 |
| CCAT | 0.60 | 67.2 | 0.43 | 90.9 | 0.77 | 45.7 | 0.78 | 44.0 | 0.98 | 17.8 |

Table 7: ***Main results for*** $98\%$***TPR on MNIST, SVHN and Cifar10.*** *While reporting results for* $99\%$*TPR in the main paper, cf. Tab. 3, reducing the TPR requirement for confidence-thresholding to* $98\%$*TPR generally improves results for both AT and CCAT, but only slightly. Again, we report Err and RErr for* $\tau = 0$ *and* $\tau@98\%$*TPR as well as ROC AUC.*

## B.4 ABLATION STUDY

Complementing Sec. 4, we include ablation studies for MNIST, SVHN and Cifar10; for our attack in Tab. 6 and training in Tab. 9.

Regarding the proposed attack, i.e., PGD-Conf, using momentum and backtracking, Tab. 6 shows that the main observations for SVHN can be transferred to MNIST and Cifar10. Only the improvement of backtracking *and* momentum over just using momentum cannot be confirmed on MNIST. Note that for fair comparison, $T$ iterations with backtracking are equivalent to $3/2T$ iterations without backtracking; which is why we include results for $T = 60$ and $T = 300$. However, the importance of using enough iterations, i.e., $T = 2000$, and zero initialization to attack CCAT is still clearly visible. Interestingly, against AT on SVHN, more iterations are also beneficial, while this is not required on MNIST and Cifar10.

Tab. 9 also reports results for CCAT with different transitions, cf. Eq. (6), and values for $\rho$. As mentioned before, on SVHN and Cifar10, power transition with $\rho_{pow} = 10$ works best; for larger $\rho$ performance stagnates. It is also important to note that the power transition does not preserve a bias towards to true label, i.e., for the maximum possible perturbation ($\|\delta\|_\infty = \epsilon$), Eq. (6) forces the network to predict a purely uniform distribution. This is in contrast to the exponential transition, where the true one-hot distribution always receives a non-zero weight. On MNIST, we found this to work considerably better.

## B.5 ANALYSIS

For further analysis, Fig. 5 shows confidence histograms for AT and CCAT on MNIST and Cifar10. The confidence histograms for CCAT reflect the expected behavior: adversarial examples are mostly successful in changing the label, which is supported by high RErr values for confidence threshold $\tau = 0$, but their confidence is pushed towards a uniform distributions. For AT, in contrast, successful adversarial examples – fewer in total – generally obtain high confidence; this results in confidence

| **MNIST:** Supplementary Results for Detection and Standard Settings | | | | | | | |
|---|---|---|---|---|---|---|---|
| ($L_\infty$ attack with $\epsilon = 0.30$ during training) | | | | | | | |
| | | **Detection Setting** $\tau$@99%TPR | | | | **Standard Setting** $\tau=0$ | |
| Attack | Training | ROC AUC | Err in % | RErr in % | $\tau$ | Err in % | RErr in % |
| $L_\infty, \epsilon = 0.30$ | AT | 0.97 | 0.0 | 1.0 | 1.0 | 0.5 | 7.2 |
| | Pang et al. (2018) | 1.00 | 0.2 | 2.5 | 2.2 | 0.5 | 100.0 |
| | CCAT | 0.99 | 0.1 | 7.7 | 1.0 | 0.5 | 100.0 |
| $L_2, \epsilon = 3$ | AT | 0.73 | 0.0 | 81.3 | 1.0 | 0.5 | 98.8 |
| | Pang et al. (2018) | 0.99 | 0.2 | 10.9 | 2.2 | 0.5 | 100.0 |
| | CCAT | 1.00 | 0.1 | 1.4 | 1.0 | 0.5 | 82.6 |

| **SVHN:** Supplementary Results for Detection and Standard Settings | | | | | | | |
|---|---|---|---|---|---|---|---|
| ($L_\infty$ attack with $\epsilon = 0.03$ during training) | | | | | | | |
| | | **Detection Setting** $\tau$@99%TPR | | | | **Standard Setting** $\tau=0$ | |
| Attack | Training | ROC AUC | Err in % | RErr in % | $\tau$ | Err in % | RErr in % |
| $L_\infty, \epsilon = 0.03$ | AT | 0.55 | 2.5 | 55.6 | 0.6 | 3.4 | 57.3 |
| | Pang et al. (2018) | 0.87 | 3.0 | 92.2 | 2.2 | 4.0 | 99.1 |
| | CCAT | 0.70 | 2.1 | 38.5 | 0.6 | 2.9 | 97.8 |
| $L_2, \epsilon = 1$ | AT | 0.26 | 2.5 | 92.0 | 0.6 | 3.4 | 92.4 |
| | Pang et al. (2018) | 0.87 | 3.0 | 81.7 | 2.2 | 4.0 | 93.9 |
| | CCAT | 0.91 | 2.1 | 18.5 | 0.6 | 2.9 | 81.8 |

| **CIFAR10:** Supplementary Results for Detection and Standard Settings | | | | | | | |
|---|---|---|---|---|---|---|---|
| ($L_\infty$ attack with $\epsilon = 0.03$ during training) | | | | | | | |
| | | **Detection Setting** $\tau$@99%TPR | | | | **Standard Setting** $\tau=0$ | |
| Attack | Training | ROC AUC | Err in % | RErr in % | $\tau$ | Err in % | RErr in % |
| $L_\infty, \epsilon = 0.03$ | AT | 0.64 | 15.1 | 62.3 | 0.3 | 16.6 | 62.7 |
| | Pang et al. (2018) | 0.70 | 19.4 | 83.2 | 2.2 | 20.0 | 99.8 |
| | CCAT | 0.60 | 8.7 | 67.9 | 0.4 | 10.1 | 96.7 |
| $L_2, \epsilon = 1$ | AT | 0.59 | 15.1 | 73.9 | 0.3 | 16.6 | 74.4 |
| | Pang et al. (2018) | 0.66 | 19.4 | 83.1 | 2.2 | 20.0 | 97.7 |
| | CCAT | 0.77 | 8.7 | 46.2 | 0.4 | 10.1 | 81.9 |

Table 8: *__Comparison with (Pang et al., 2018) on MNIST, SVHN and Cifar10.__ We report "standard" Err and RErr, i.e., $\tau = 0$ ("Standard Setting"), as well as their confidence thresholded variants for $\tau$@99%TPR ("Detection Setting"), cf. Sec. 3. On MNIST their method is competitive regarding RErr after confidence-thresholding, even outperforming AT when it comes to the generalization to $L_2$ attacks. However, on SVHN and Cifar10, the approach cannot be considered robust anymore regarding our rigorous evaluation; this might also be due to the fact that no adversarial examples are used during training.*

thresholding being not effective for AT. This behavior, however, is less pronounced on MNIST. Here, the exponential transition results in adversarial examples with confidence slightly higher than uniform confidence, i.e., 0.1. This might be the result of preserving a bias towards the true label through Eq. (6). In fact, for lower $\rho$, we found that this behavior is pronounced, until, for very small $\rho$, the behavior of AT is obtained.

In Fig. 7, we plot the probabilities for all ten classes along an adversarial direction. We note that these directions do not necessarily correspond to successful adversarial examples. Instead, we chose the first 10 test examples on SVHN. The adversarial examples were obtained using our $L_\infty$ PGD-Conf attack with $T = 2000$ iterations and zero initialization for $\epsilon = 0.03$. For AT, we usually observe a change in predictions along these directions; some occur within $\|\delta\|_\infty \leq \epsilon$, corresponding to successful adversarial examples, some occur for $\|\delta\|_\infty > \epsilon$, corresponding to unsuccessful adversarial examples (within $\epsilon$). However, AT always assigns high confidence. Thus, when allowing larger adversarial perturbations at test time, robustness of AT reduces significantly. For CCAT, in

contrast, there are only few such cases; more often, the model achieves a near uniform prediction for small $\|\delta\|_\infty$ and extrapolates this behavior beyond the $\epsilon$-ball used for training. On SVHN, this behavior successfully allows to generalize the robustness to larger adversarial perturbations. Furthermore, these plots illustrate why using more iterations at test time, and using techniques such as momentum and backtracking, are necessary to find adversarial examples as the objective becomes more complex compared to AT.

*Fig. 8 shows concrete adversarial examples against AT and CCAT on MNIST and SVHN. Among the first 200 test examples, we picked the first six examples where AT is mistaken and (non-overlapping) six examples where CCAT has difficulties, i.e., assigns high confidence. On SVHN, for example, CCAT is able to detect adversarial examples with low confidence where AT assigns incorrect classes. On MNIST, in contrast, CCAT, cannot recover mistakes made by AT. Similarly, there exist examples where AT assigns the correct class while CCAT assigns the wrong class with very high confidence. On MNIST, we additionally found adversarial examples against CCAT to represent more "localized" changes in the $L_\infty$ norm compared to AT. Overall, however, we did not find any systematic difference between strong adversarial against AT and CCAT.*

*In Fig. 9, on MNIST, we additionally illustrate the advantage of CCAT with respect to the toy example in Proposition 1. Here, we consider the case where the $\epsilon$-balls of two training or test examples (in different classes) overlap. As we show in Proposition 1, adversarial training is not able to handle such cases, resulting in the trade-off between accuracy in robustness reported in the literature (Tsipras et al., 2018; Stutz et al., 2019; Raghunathan et al., 2019; Zhang et al., 2019). This is because adversarial training enforces high-confidence predictions on both $\epsilon$-balls (corresponding to different classes), resulting in an obvious conflict. CCAT, in contrast, enforces uniform predictions throughout the largest parts of both $\epsilon$-balls, resolving the conflict. Finally, Fig. 9 also shows that CCAT is indeed able to extrapolate the uniform predictions beyond the $\epsilon$-ball used during training.*

## B.6 RESULTS

In the following, we present and discuss complementary results corresponding to the main results of our paper as presented in Tab. 3 and 4. To this end, we consider requiring only 98% TPR instead of 99% TPR, and most importantly, break down our analysis by the different white-box and black-box attacks used.

**Main results for** 98% **TPR:** Tab. 7 reports results (corresponding to Tab. 3) requiring only 98%TPR. This implies, that compared to 99%TPR, up to 1% more correctly classified test examples can be rejected. For relatively simple tasks such as MNIST and SVHN, where Err values are low, this is a significant "sacrifice". However, as can be seen, robustness in terms of RErr only improves slightly. We found that the same holds for 95%TPR, *however, rejecting more than 2% correctly classified examples seems prohibitive large for the considered datasets.*

**Per-Attack Results:** Finally Tab. 10, 11 and 12 we break down the results of Tab. 3 regarding the used attacks. For simplicity we focus on PGD-CE and PGD-Conf while reporting the used black-box attacks together, i.e., taking the worst-case adversarial examples across all black-box attacks. *For comparison, we also include non-thresholded Err and RErr for the results from the main paper.* On MNIST, where AT performs very well in practice, it is striking that for $4/3\epsilon = 0.4$ even black-box attacks are able to reduce robustness completely, resulting in high RErr. This observation also transfers to SVHN and Cifar10. For CCAT, black-box attacks are only effective on Cifar10, where they result in roughly 87% RErr with $\tau$@99%TPR. *For the $L_2$, $L_1$ and $L_0$ attacks we can make similar observations, except on MNIST, where AT seems similarly robust against $L_1$ and $L_0$ attacks as CCAT. Across all $L_p$ norms,* it can also be seen that PGD-CE performs significantly worse against our CCAT compared to AT, which shows that it is essential to optimize the right objective *to evaluate the robustness of defenses and adversarially trained models*, i.e., maximize confidence against CCAT

*Results on Corrupted MNIST/Cifar10: We also conducted experiments on MNIST-C (Mu & Gilmer, 2019) and Cifar10-C (Hendrycks & Dietterich, 2019). These datasets are variants of MNIST and Cifar10 that contain common perturbations of the original images obtained from various types of noise, blur or transformations; examples include zoom or motion blue, Gaussian and shot noise, rotations, translations and shear. Tab. 14 presents the per-corruption results on MNIST-C and Cifar10-C. Here,* `all` *includes all corruptions and* `mean` *reports the average results across*

*all corruptions; note that, due to the thresholding, leaving different numbers of corrupted examples after detection, the distinction between `all` and `mean` is meaningful. Striking is the performance of CCAT on noise corruptions such as `gaussian_noise` or `shot_noise`. Here, CCAT is able to detect $100\%$ of the corrupted examples (cf. the true negative rate (TNR) in Tab. 14), resulting in a thresholded Err of $0\%$. This is in stark contrast to AT, exhibiting a Err of roughly $15\%$ after detection on Cifar10-C. On the remaining corruptions, CCAT is able to perform slightly better than AT, which is often due to higher detection rate, i.e., higher ROC AUC. On, Cifar10, the generally lower Err of CCAT also contributes to the results. Overall, this illustrates that CCAT is able to preserve the inductive bias of predicting near-uniform distribution on noise similar to $L_\infty$ adversarial examples as seen during training.*

***Comparison with (Pang et al., 2018):** We also compare AT and CCAT with the approach by Pang et al. (2018). We implemented the reverse cross-entropy error in PyTorch following the official code[1]. The reverse cross-entropy loss encourages the non-maximum predictions (i.e., in the case of correct classification, the probabilities of the "other" classes) to be uniform. Then, adversarial examples are detected based on statistics of the predicted distributions. We used the non-maximum element entropy (nonME) detector instead of the K-density detector (Feinman et al., 2017) which has been shown to be ineffective against adaptive attacks in (Carlini & Wagner, 2017a). We note that (Pang et al., 2018) do not train on adversarial examples as AT or CCAT. Our evaluation metrics need not be changed; however, thresholding is done on the nonME detector instead of on the confidence, as for CCAT. In Tab. 8, it can be seen, that the approach works considerably well on MNIST, regarding both $L_\infty$ and $L_2$ attacks. Again, we report the worst-case results across PGD-CE, PGD-Conf and the considered black-box attacks. However, on SVHN and Cifar10, we are able to break to robustness of the approach, reaching $92\%$ (thresholded) RErr on SVHN and $99\%$ RTE on Cifar10. On SVHN, the models performs slightly better against $L_2$ attacks, with $81\%$, but can not be considered robust anymore.*

---

[1] https://github.com/P2333/Reverse-Cross-Entropy

| MNIST: Training Ablation for Detection and Standard Settings ($L_\infty$ attack with $\epsilon = 0.30$ during training) | | | | | | |
|---|---|---|---|---|---|---|
| | Detection Setting $\tau$@99%TPR | | | | Standard Setting $\tau = 0$ | |
| | ROC AUC | Err in % | RErr in % | $\tau$ | Err in % | RErr in % |
| Normal | 0.34 | 0.1 | 100.0 | 1.0 | 0.4 | 100.0 |
| AT | 0.97 | 0.0 | 0.4 | 1.0 | 0.5 | 5.6 |
| AT Conf | 0.98 | 0.1 | 1.1 | 1.0 | 0.5 | 6.0 |
| CCAT, $\rho_{exp} = 3$ | 0.99 | 0.1 | 11.9 | 1.0 | 0.4 | 88.1 |
| CCAT, $\rho_{exp} = 4$ | 0.98 | 0.1 | 10.4 | 1.0 | 0.5 | 95.2 |
| CCAT, $\rho_{exp} = 5$ | 0.99 | 0.1 | 10.8 | 1.0 | 0.4 | 88.7 |
| CCAT, $\rho_{exp} = 6$ | 0.99 | 0.1 | 7.8 | 1.0 | 0.4 | 83.5 |
| CCAT, $\rho_{exp} = 7$ | 0.99 | 0.1 | 5.7 | 1.0 | 0.5 | 74.7 |
| CCAT, $\rho_{exp} = 8$ | 0.98 | 0.1 | 13.3 | 1.0 | 0.3 | 94.3 |
| CCAT, $\rho_{exp} = 9$ | 0.99 | 0.1 | 11.0 | 1.0 | 0.3 | 86.8 |
| CCAT, $\rho_{exp} = 10$ | 0.99 | 0.1 | 14.3 | 1.0 | 0.4 | 97.1 |
| CCAT, $\rho_{pow} = 10$ | 0.95 | 0.1 | 17.3 | 1.0 | 0.3 | 64.8 |

| SVHN: Training Ablation for Detection and Standard Settings ($L_\infty$ attack with $\epsilon = 0.03$ during training) | | | | | | |
|---|---|---|---|---|---|---|
| | Detection Setting $\tau$@99%TPR | | | | Standard Setting $\tau = 0$ | |
| | ROC AUC | Err in % | RErr in % | $\tau$ | Err in % | RErr in % |
| Normal | 0.17 | 2.6 | 99.9 | 0.8 | 3.6 | 99.9 |
| AT | 0.55 | 2.5 | 54.9 | 0.6 | 3.4 | 56.9 |
| AT Conf | 0.61 | 2.8 | 52.5 | 0.6 | 3.7 | 58.7 |
| CCAT, $\rho_{pow} = 1$ | 0.74 | 2.2 | 43.0 | 0.3 | 2.7 | 82.4 |
| CCAT, $\rho_{pow} = 2$ | 0.68 | 2.1 | 44.2 | 0.5 | 2.9 | 79.6 |
| CCAT, $\rho_{pow} = 4$ | 0.68 | 1.8 | 35.8 | 0.6 | 2.7 | 80.4 |
| CCAT, $\rho_{pow} = 6$ | 0.64 | 1.8 | 32.8 | 0.7 | 2.9 | 72.1 |
| CCAT, $\rho_{pow} = 8$ | 0.63 | 2.2 | 42.3 | 0.6 | 2.9 | 84.6 |
| CCAT, $\rho_{pow} = 10$ | 0.67 | 2.1 | 38.5 | 0.6 | 2.9 | 91.0 |
| CCAT, $\rho_{pow} = 12$ | 0.67 | 1.9 | 36.3 | 0.6 | 2.8 | 81.8 |
| CCAT, $\rho_{exp} = 7$ | 0.66 | 2.0 | 54.7 | 0.5 | 2.9 | 73.1 |

| CIFAR10: Training Ablation for Detection and Standard Settings ($L_\infty$ attack with $\epsilon = 0.03$ during training) | | | | | | |
|---|---|---|---|---|---|---|
| | Detection Setting $\tau$@99%TPR | | | | Standard Setting $\tau = 0$ | |
| | ROC AUC | Err in % | RErr in % | $\tau$ | Err in % | RErr in % |
| Normal | 0.20 | 7.4 | 100.0 | 0.6 | 8.3 | 100.0 |
| AT | 0.65 | 15.1 | 60.9 | 0.3 | 16.6 | 61.3 |
| AT Conf | 0.63 | 15.1 | 61.5 | 0.4 | 16.1 | 61.7 |
| CCAT, $\rho_{pow} = 1$ | 0.63 | 8.7 | 72.4 | 0.3 | 9.7 | 95.3 |
| CCAT, $\rho_{pow} = 2$ | 0.60 | 8.4 | 70.6 | 0.4 | 9.7 | 95.1 |
| CCAT, $\rho_{pow} = 4$ | 0.61 | 8.6 | 66.3 | 0.4 | 9.8 | 93.5 |
| CCAT, $\rho_{pow} = 6$ | 0.54 | 8.0 | 69.8 | 0.4 | 9.2 | 94.1 |
| CCAT, $\rho_{pow} = 8$ | 0.58 | 8.5 | 65.3 | 0.4 | 9.4 | 93.2 |
| CCAT, $\rho_{pow} = 10$ | 0.60 | 8.7 | 63.0 | 0.4 | 10.1 | 95.0 |
| CCAT, $\rho_{pow} = 12$ | 0.62 | 9.4 | 63.0 | 0.3 | 10.1 | 96.6 |
| CCAT, $\rho_{exp} = 7$ | 0.66 | 11.7 | 68.6 | 0.2 | 13.2 | 77.4 |

Table 9: **Training ablation studies on MNIST, SVHN and Cifar10.** We report results for different $\rho$ and transitions, cf. Eq. (6). We report RErr and Err with confidence threshold $\tau = 0$ ("Standard Setting") and $\tau$@99%TPR as well as ROC AUC ("Detection Setting"). The models are tested against our $L_\infty$ PGD-Conf attack with $T = 2000$ iterations and zero as well as random initialization, *as discussed in Sec. 4.1.* On MNIST, the exponential transition, especially $\rho_{exp} = 7$ performs best; on Cifar10, the power transition with $\rho_{pow} = 10$ works best – performance stagnates for $\rho_{pow} > 10$. On SVHN, we also use $\rho_{pow} = 10$, although $\rho_{pow} = 6$ shows better results. However, against larger $\epsilon$-balls, we found that $\rho_{pow} = 10$ works significantly better.

| MNIST: Supplementary Results for Detection and Standard Settings | | | | | | | |
|---|---|---|---|---|---|---|---|
| ($L_\infty, \epsilon = 0.30$ during training) | | | | | | | |
| | | **Detection Setting** $\tau$@99%TPR | | | | **Standard Setting** $\tau{=}0$ | |
| Attack | Training | ROC AUC | Err in % | RErr in % | $\tau$ | Err in % | RErr in % |
| Worst-Case ($L_\infty, \epsilon = 0.30$) | AT | 0.97 | 0.0 | 1.0 | 1.0 | 0.5 | 7.2 |
| | CCAT, $\rho_{\mathrm{exp}}{=}7$ | 0.99 | 0.1 | 7.7 | 1.0 | 0.5 | 100.0 |
| | CCAT, $\rho_{\mathrm{pow}}{=}10$ | 0.96 | 0.1 | 21.9 | 1.0 | 0.3 | 90.5 |
| PGD Conf ($L_\infty, \epsilon = 0.30$) | AT | 0.97 | 0.0 | 0.4 | 1.0 | 0.5 | 5.6 |
| | CCAT, $\rho_{\mathrm{exp}}{=}7$ | 0.99 | 0.1 | 5.7 | 1.0 | 0.5 | 74.7 |
| | CCAT, $\rho_{\mathrm{pow}}{=}10$ | 0.95 | 0.1 | 17.3 | 1.0 | 0.3 | 64.8 |
| PGD CE ($L_\infty, \epsilon = 0.30$) | AT | 0.97 | 0.0 | 0.8 | 1.0 | 0.5 | 6.7 |
| | CCAT, $\rho_{\mathrm{exp}}{=}7$ | 1.00 | 0.1 | 4.3 | 1.0 | 0.5 | 100.0 |
| | CCAT, $\rho_{\mathrm{pow}}{=}10$ | 1.00 | 0.1 | 0.1 | 1.0 | 0.3 | 100.0 |
| Black-Box ($L_\infty, \epsilon = 0.30$) | AT | 0.98 | 0.0 | 1.0 | 1.0 | 0.5 | 7.2 |
| | CCAT, $\rho_{\mathrm{exp}}{=}7$ | 1.00 | 0.1 | 0.4 | 1.0 | 0.5 | 100.0 |
| | CCAT, $\rho_{\mathrm{pow}}{=}10$ | 1.00 | 0.1 | 7.4 | 1.0 | 0.3 | 88.6 |
| Worst-Case ($L_\infty, \epsilon = 0.40$) | AT | 0.20 | 0.0 | 100.0 | 1.0 | 0.5 | 100.0 |
| | CCAT, $\rho_{\mathrm{exp}}{=}7$ | 0.94 | 0.1 | 40.0 | 1.0 | 0.5 | 100.0 |
| | CCAT, $\rho_{\mathrm{pow}}{=}10$ | 0.94 | 0.1 | 29.0 | 1.0 | 0.3 | 96.9 |
| PGD Conf ($L_\infty, \epsilon = 0.40$) | AT | 0.36 | 0.0 | 97.8 | 1.0 | 0.5 | 99.8 |
| | CCAT, $\rho_{\mathrm{exp}}{=}7$ | 0.96 | 0.1 | 15.4 | 1.0 | 0.5 | 97.1 |
| | CCAT, $\rho_{\mathrm{pow}}{=}10$ | 0.92 | 0.1 | 20.6 | 1.0 | 0.3 | 72.5 |
| PGD CE ($L_\infty, \epsilon = 0.40$) | AT | 0.20 | 0.0 | 100.0 | 1.0 | 0.5 | 100.0 |
| | CCAT, $\rho_{\mathrm{exp}}{=}7$ | 0.97 | 0.1 | 29.6 | 1.0 | 0.5 | 100.0 |
| | CCAT, $\rho_{\mathrm{pow}}{=}10$ | 1.00 | 0.1 | 2.5 | 1.0 | 0.3 | 100.0 |
| Black-Box ($L_\infty, \epsilon = 0.40$) | AT | 0.23 | 0.0 | 100.0 | 1.0 | 0.5 | 100.0 |
| | CCAT, $\rho_{\mathrm{exp}}{=}7$ | 0.99 | 0.1 | 3.9 | 1.0 | 0.5 | 100.0 |
| | CCAT, $\rho_{\mathrm{pow}}{=}10$ | 0.99 | 0.1 | 10.7 | 1.0 | 0.3 | 91.8 |
| Worst-Case ($L_2, \epsilon = 3$) | AT | 0.73 | 0.0 | 81.3 | 1.0 | 0.5 | 98.8 |
| | CCAT, $\rho_{\mathrm{exp}}{=}7$ | 1.00 | 0.1 | 1.4 | 1.0 | 0.5 | 82.6 |
| | CCAT, $\rho_{\mathrm{pow}}{=}10$ | 1.00 | 0.1 | 0.1 | 1.0 | 0.3 | 22.1 |
| PGD Conf ($L_2, \epsilon = 3$) | AT | 0.98 | 0.0 | 0.2 | 1.0 | 0.5 | 3.4 |
| | CCAT, $\rho_{\mathrm{exp}}{=}7$ | 1.00 | 0.1 | 0.0 | 1.0 | 0.5 | 4.4 |
| | CCAT, $\rho_{\mathrm{pow}}{=}10$ | 1.00 | 0.1 | 0.1 | 1.0 | 0.3 | 9.5 |
| PGD CE ($L_2, \epsilon = 3$) | AT | 0.93 | 0.0 | 11.5 | 1.0 | 0.5 | 29.2 |
| | CCAT, $\rho_{\mathrm{exp}}{=}7$ | 1.00 | 0.1 | 0.9 | 1.0 | 0.5 | 82.4 |
| | CCAT, $\rho_{\mathrm{pow}}{=}10$ | 1.00 | 0.1 | 0.1 | 1.0 | 0.3 | 100.0 |
| Black-Box ($L_2, \epsilon = 3$) | AT | 0.73 | 0.0 | 80.1 | 1.0 | 0.5 | 98.7 |
| | CCAT, $\rho_{\mathrm{exp}}{=}7$ | 1.00 | 0.1 | 0.6 | 1.0 | 0.5 | 38.4 |
| | CCAT, $\rho_{\mathrm{pow}}{=}10$ | 1.00 | 0.1 | 0.1 | 1.0 | 0.3 | 100.0 |
| Worst-Case ($L_1, \epsilon = 10.00$) | AT | 0.93 | 0.0 | 5.2 | 1.0 | 0.5 | 18.0 |
| | CCAT, $\rho_{\mathrm{exp}}{=}7$ | 0.96 | 0.1 | 14.7 | 1.0 | 0.5 | 35.8 |
| | CCAT, $\rho_{\mathrm{pow}}{=}10$ | 1.00 | 0.1 | 1.6 | 1.0 | 0.3 | 17.7 |
| PGD Conf ($L_1, \epsilon = 10.00$) | AT | 0.93 | 0.0 | 4.2 | 1.0 | 0.5 | 14.8 |
| | CCAT, $\rho_{\mathrm{exp}}{=}7$ | 0.96 | 0.1 | 14.5 | 1.0 | 0.5 | 35.7 |
| | CCAT, $\rho_{\mathrm{pow}}{=}10$ | 1.00 | 0.1 | 1.6 | 1.0 | 0.3 | 17.7 |
| PGD CE ($L_1, \epsilon = 10.00$) | AT | 0.96 | 0.0 | 3.8 | 1.0 | 0.5 | 15.5 |
| | CCAT, $\rho_{\mathrm{exp}}{=}7$ | 0.98 | 0.1 | 4.2 | 1.0 | 0.5 | 12.1 |
| | CCAT, $\rho_{\mathrm{pow}}{=}10$ | 1.00 | 0.1 | 0.1 | 1.0 | 0.3 | 99.7 |
| Worst-Case ($L_0, \epsilon = 15$) | AT | 0.99 | 0.0 | 2.5 | 1.0 | 0.5 | 93.9 |
| | CCAT, $\rho_{\mathrm{exp}}{=}7$ | 0.99 | 0.1 | 7.8 | 1.0 | 0.5 | 65.9 |
| | CCAT, $\rho_{\mathrm{pow}}{=}10$ | 0.99 | 0.1 | 7.8 | 1.0 | 0.3 | 88.8 |
| PGD Conf ($L_0, \epsilon = 15$) | AT | 0.98 | 0.0 | 0.5 | 1.0 | 0.5 | 5.2 |
| | CCAT, $\rho_{\mathrm{exp}}{=}7$ | 0.99 | 0.1 | 4.1 | 1.0 | 0.5 | 18.3 |
| | CCAT, $\rho_{\mathrm{pow}}{=}10$ | 0.99 | 0.1 | 3.7 | 1.0 | 0.3 | 13.4 |
| PGD CE ($L_0, \epsilon = 15$) | AT | 0.98 | 0.0 | 2.4 | 1.0 | 0.5 | 17.7 |
| | CCAT, $\rho_{\mathrm{exp}}{=}7$ | 0.98 | 0.1 | 6.2 | 1.0 | 0.5 | 18.6 |
| | CCAT, $\rho_{\mathrm{pow}}{=}10$ | 0.98 | 0.1 | 5.4 | 1.0 | 0.3 | 16.5 |
| Black-Box ($L_0, \epsilon = 15$) | AT | 1.00 | 0.0 | 0.0 | 1.0 | 0.5 | 93.9 |
| | CCAT, $\rho_{\mathrm{exp}}{=}7$ | 1.00 | 0.1 | 0.1 | 1.0 | 0.5 | 65.7 |
| | CCAT, $\rho_{\mathrm{pow}}{=}10$ | 1.00 | 0.1 | 1.4 | 1.0 | 0.3 | 88.9 |

Table 10: **Per-attack results on MNIST.** Per-attack results considering PGD-CE, as in Madry et al. (2018), our PGD-Conf and the remaining black-box attacks *for all threat models, i.e., $L_\infty$, $L_2$, $L_1$ and $L_0$*, see text. The used $\epsilon$ values are reported in the left-most column. For the black-box attacks, we take the per-example worst-case across all black-box attacks.

| SVHN: Supplementary Results for Detection and Standard Settings $(L_\infty, \epsilon = 0.03$ during training) | | | | | | | |
|---|---|---|---|---|---|---|---|
| | | **Detection Setting** $\tau$@99%TPR | | | | **Standard Setting** $\tau=0$ | |
| Attack | Training | ROC AUC | Err in % | RErr in % | $\tau$ | Err in % | RErr in % |
| Worst-Case $(L_\infty, \epsilon = 0.03)$ | AT | 0.55 | 2.5 | 55.6 | 0.6 | 3.4 | 57.3 |
| | CCAT | 0.70 | 2.1 | 38.5 | 0.6 | 2.9 | 97.8 |
| PGD Conf $(L_\infty, \epsilon = 0.03)$ | AT | 0.55 | 2.5 | 54.9 | 0.6 | 3.4 | 56.9 |
| | CCAT | 0.67 | 2.1 | 38.5 | 0.6 | 2.9 | 91.0 |
| PGD CE $(L_\infty, \epsilon = 0.03)$ | AT | 0.68 | 2.5 | 43.2 | 0.6 | 3.4 | 50.7 |
| | CCAT | 1.00 | 2.1 | 2.6 | 0.6 | 2.9 | 94.9 |
| Black-Box $(L_\infty, \epsilon = 0.03)$ | AT | 0.95 | 2.5 | 30.8 | 0.6 | 3.4 | 46.2 |
| | CCAT | 1.00 | 2.1 | 6.3 | 0.6 | 2.9 | 79.5 |
| Worst-Case $(L_\infty, \epsilon = 0.06)$ | AT | 0.32 | 2.5 | 88.3 | 0.6 | 3.4 | 89.0 |
| | CCAT | 0.70 | 2.1 | 46.0 | 0.6 | 2.9 | 99.8 |
| PGD Conf $(L_\infty, \epsilon = 0.06)$ | AT | 0.32 | 2.5 | 84.7 | 0.6 | 3.4 | 86.1 |
| | CCAT | 0.70 | 2.1 | 36.8 | 0.6 | 2.9 | 98.7 |
| PGD CE $(L_\infty, \epsilon = 0.06)$ | AT | 0.66 | 2.5 | 88.0 | 0.6 | 3.4 | 88.9 |
| | CCAT | 0.99 | 2.1 | 17.2 | 0.6 | 2.9 | 100.0 |
| Black-Box $(L_\infty, \epsilon = 0.06)$ | AT | 0.78 | 2.5 | 82.3 | 0.6 | 3.4 | 84.0 |
| | CCAT | 1.00 | 2.1 | 4.8 | 0.6 | 2.9 | 82.3 |
| Worst-Case$(L_2, \epsilon = 1)$ | AT | 0.26 | 2.5 | 92.0 | 0.6 | 3.4 | 92.4 |
| | CCAT | 0.91 | 2.1 | 18.5 | 0.6 | 2.9 | 81.8 |
| PGD Conf $(L_2, \epsilon = 1)$ | AT | 0.50 | 2.5 | 77.1 | 0.6 | 3.4 | 78.7 |
| | CCAT | 0.90 | 2.1 | 18.4 | 0.6 | 2.9 | 74.4 |
| PGD CE $(L_2, \epsilon = 1)$ | AT | 0.27 | 2.5 | 92.0 | 0.6 | 3.4 | 92.4 |
| | CCAT | 0.99 | 2.1 | 3.7 | 0.6 | 2.9 | 100.0 |
| Black-Box $(L_2, \epsilon = 1)$ | AT | 0.98 | 2.5 | 13.7 | 0.6 | 3.4 | 29.8 |
| | CCAT | 1.00 | 2.1 | 2.6 | 0.6 | 2.9 | 100.0 |
| Worst-Case $(L_1, \epsilon = 7.85)$ | AT | 0.34 | 2.5 | 91.9 | 0.6 | 3.4 | 92.5 |
| | CCAT | 0.89 | 2.1 | 20.8 | 0.6 | 2.9 | 63.5 |
| PGD Conf $(L_1, \epsilon = 7.85)$ | AT | 0.37 | 2.5 | 87.9 | 0.6 | 3.4 | 88.8 |
| | CCAT | 0.92 | 2.1 | 13.4 | 0.6 | 2.9 | 50.3 |
| PGD CE $(L_1, \epsilon = 7.85)$ | AT | 0.51 | 2.5 | 88.9 | 0.6 | 3.4 | 89.6 |
| | CCAT | 0.99 | 2.1 | 4.8 | 0.6 | 2.9 | 100.0 |
| Worst-Case $(L_0, \epsilon = 10)$ | AT | 0.90 | 2.5 | 73.4 | 0.6 | 3.4 | 89.2 |
| | CCAT | 1.00 | 2.1 | 2.7 | 0.6 | 2.9 | 77.1 |
| PGD Conf $(L_0, \epsilon = 10)$ | AT | 0.90 | 2.5 | 51.3 | 0.6 | 3.4 | 59.7 |
| | CCAT | 1.00 | 2.1 | 2.7 | 0.6 | 2.9 | 63.9 |
| PGD CE $(L_0, \epsilon = 10)$ | AT | 0.89 | 2.5 | 60.8 | 0.6 | 3.4 | 68.9 |
| | CCAT | 1.00 | 2.1 | 2.6 | 0.6 | 2.9 | 96.9 |
| Black-Box $(L_0, \epsilon = 10)$ | AT | 0.98 | 2.5 | 35.0 | 0.6 | 3.4 | 86.7 |
| | CCAT | 1.00 | 2.1 | 2.6 | 0.6 | 2.9 | 100.0 |

Table 11: **Per-attack results on SVHN.** Per-attack results considering PGD-CE, as in Madry et al. (2018), our PGD-Conf and the remaining black-box attacks *for all threat models, i.e., $L_\infty$, $L_2$, $L_1$ and $L_0$,* see text. The used $\epsilon$ values are reported in the left-most column. For the black-box attacks, we take the per-example worst-case across all black-box attacks.

| CIFAR10: Supplementary Results for Detection and Standard Settings | | | | | | | |
|:---:|:---:|:---:|:---:|:---:|:---:|:---:|:---:|
| ($L_\infty$, $\epsilon = 0.03$ during training) | | | | | | | |
| | | **Detection Setting** $\tau$@99%TPR | | | | **Standard Setting** $\tau$=0 | |
| Attack | Training | ROC AUC | Err in % | RErr in % | $\tau$ | Err in % | RErr in % |
| Worst-Case ($L_\infty$, $\epsilon = 0.03$) | AT | 0.64 | 15.1 | 62.3 | 0.3 | 16.6 | 62.7 |
| | CCAT | 0.60 | 8.7 | 67.9 | 0.4 | 10.1 | 96.7 |
| PGD Conf ($L_\infty$, $\epsilon = 0.03$) | AT | 0.65 | 15.1 | 60.9 | 0.3 | 16.6 | 61.3 |
| | CCAT | 0.60 | 8.7 | 63.0 | 0.4 | 10.1 | 95.0 |
| PGD CE ($L_\infty$, $\epsilon = 0.03$) | AT | 0.67 | 15.1 | 61.4 | 0.3 | 16.6 | 62.3 |
| | CCAT | 0.99 | 8.7 | 9.8 | 0.4 | 10.1 | 100.0 |
| Black-Box ($L_\infty$, $\epsilon = 0.03$) | AT | 0.70 | 15.1 | 56.9 | 0.3 | 16.6 | 57.3 |
| | CCAT | 0.90 | 8.7 | 49.3 | 0.4 | 10.1 | 96.4 |
| Worst-Case ($L_\infty$, $\epsilon = 0.06$) | AT | 0.35 | 15.1 | 93.6 | 0.3 | 16.6 | 93.7 |
| | CCAT | 0.43 | 8.7 | 91.5 | 0.4 | 10.1 | 99.2 |
| PGD Conf ($L_\infty$, $\epsilon = 0.06$) | AT | 0.37 | 15.1 | 92.1 | 0.3 | 16.6 | 92.2 |
| | CCAT | 0.49 | 8.7 | 66.8 | 0.4 | 10.1 | 97.3 |
| PGD CE ($L_\infty$, $\epsilon = 0.06$) | AT | 0.40 | 15.1 | 93.6 | 0.3 | 16.6 | 93.7 |
| | CCAT | 0.98 | 8.7 | 10.4 | 0.4 | 10.1 | 100.0 |
| Black-Box ($L_\infty$, $\epsilon = 0.06$) | AT | 0.50 | 15.1 | 87.1 | 0.3 | 16.6 | 87.2 |
| | CCAT | 0.78 | 8.7 | 87.0 | 0.4 | 10.1 | 99.5 |
| Worst-Case($L_2$, $\epsilon = 1$) | AT | 0.59 | 15.1 | 73.9 | 0.3 | 16.6 | 74.4 |
| | CCAT | 0.77 | 8.7 | 46.2 | 0.4 | 10.1 | 81.9 |
| PGD Conf ($L_2$, $\epsilon = 1$) | AT | 0.63 | 15.1 | 64.9 | 0.3 | 16.6 | 65.3 |
| | CCAT | 0.78 | 8.7 | 45.2 | 0.4 | 10.1 | 80.9 |
| PGD CE ($L_2$, $\epsilon = 1$) | AT | 0.61 | 15.1 | 73.9 | 0.3 | 16.6 | 74.6 |
| | CCAT | 0.95 | 8.7 | 18.9 | 0.4 | 10.1 | 100.0 |
| Black-Box ($L_2$, $\epsilon = 1$) | AT | 0.81 | 15.1 | 35.8 | 0.3 | 16.6 | 36.9 |
| | CCAT | 1.00 | 8.7 | 8.8 | 0.4 | 10.1 | 100.0 |
| Worst-Case ($L_1$, $\epsilon = 7.85$) | AT | 0.61 | 15.1 | 68.0 | 0.3 | 16.6 | 68.3 |
| | CCAT | 0.78 | 8.7 | 45.2 | 0.4 | 10.1 | 75.7 |
| PGD Conf ($L_1$, $\epsilon = 7.85$) | AT | 0.61 | 15.1 | 66.8 | 0.3 | 16.6 | 67.1 |
| | CCAT | 0.84 | 8.7 | 35.7 | 0.4 | 10.1 | 73.5 |
| PGD CE ($L_1$, $\epsilon = 7.85$) | AT | 0.71 | 15.1 | 58.5 | 0.3 | 16.6 | 60.5 |
| | CCAT | 0.96 | 8.7 | 17.4 | 0.4 | 10.1 | 100.0 |
| Worst-Case ($L_0$, $\epsilon = 10$) | AT | 0.80 | 15.1 | 74.1 | 0.3 | 16.6 | 75.9 |
| | CCAT | 0.98 | 8.7 | 20.9 | 0.4 | 10.1 | 55.7 |
| PGD Conf ($L_0$, $\epsilon = 10$) | AT | 0.74 | 15.1 | 44.3 | 0.3 | 16.6 | 45.1 |
| | CCAT | 0.98 | 8.7 | 12.5 | 0.4 | 10.1 | 34.8 |
| PGD CE ($L_0$, $\epsilon = 10$) | AT | 0.76 | 15.1 | 49.4 | 0.3 | 16.6 | 51.4 |
| | CCAT | 1.00 | 8.7 | 7.6 | 0.4 | 10.1 | 79.3 |
| Black-Box ($L_0$, $\epsilon = 10$) | AT | 0.91 | 15.1 | 69.6 | 0.3 | 16.6 | 75.9 |
| | CCAT | 0.99 | 8.7 | 17.8 | 0.4 | 10.1 | 99.1 |

Table 12: **Per-attack $L_\infty$ and $L_2$ results on Cifar10.** Per-attack results considering PGD-CE, as in Madry et al. (2018), our PGD-Conf and the remaining black-box attacks *for all threat models, i.e., $L_\infty$, $L_2$, $L_1$ and $L_0$,* see text. The used $\epsilon$ values are reported in the left-most column. For the black-box attacks, we take the per-example worst-case across all black-box attacks.

| MNIST: Supplementary Results for Detection and Standard Settings | | | | | | | |
|---|---|---|---|---|---|---|---|
| ($L_\infty$ attack with $\epsilon = 0.30$ during training) | | | | | | | |
| | | Detection Setting $\tau$@99%TPR | | | | | Standard Setting $\tau$=0 |
| Corruption | Training | ROC AUC | FPR in % | TNR in % | Err in % | $\tau$ | Err in % |
| all | Normal | 0.75 | 82.8 | 17.2 | 31.9 | 1.0 | 36.4 |
| | AT | 0.80 | 59.0 | 41.0 | 4.0 | 1.0 | 26.9 |
| | CCAT | 0.95 | 40.3 | 59.7 | 4.5 | 1.0 | 25.1 |
| mean | Normal | 0.75 | 82.8 | 17.2 | 32.8 | 1.0 | 36.3 |
| | AT | 0.80 | 59.0 | 41.0 | 12.6 | 1.0 | 26.9 |
| | CCAT | 0.95 | 40.4 | 59.7 | 4.0 | 1.0 | 25.1 |
| brightness | Normal | 0.36 | 100.0 | 0.0 | 90.2 | 1.0 | 90.2 |
| | AT | 0.99 | 0.9 | 99.1 | 36.4 | 1.0 | 84.3 |
| | CCAT | 1.00 | 0.0 | 100.0 | 0.0 | 1.0 | 88.3 |
| canny_edges | Normal | 0.91 | 62.4 | 37.6 | 34.8 | 1.0 | 45.6 |
| | AT | 0.96 | 28.8 | 71.2 | 33.4 | 1.0 | 53.2 |
| | CCAT | 0.97 | 45.1 | 54.9 | 47.0 | 1.0 | 51.5 |
| dotted_line | Normal | 0.76 | 85.4 | 14.6 | 2.4 | 1.0 | 7.6 |
| | AT | 0.77 | 72.8 | 27.2 | 0.9 | 1.0 | 7.9 |
| | CCAT | 0.92 | 74.2 | 25.8 | 1.8 | 1.0 | 7.6 |
| fog | Normal | 0.38 | 99.9 | 0.1 | 90.2 | 1.0 | 90.2 |
| | AT | 0.90 | 29.7 | 70.3 | 10.4 | 1.0 | 59.0 |
| | CCAT | 1.00 | 0.0 | 100.0 | 0.0 | 1.0 | 65.0 |
| glass_blur | Normal | 0.87 | 71.6 | 28.4 | 57.2 | 1.0 | 56.2 |
| | AT | 0.82 | 67.4 | 32.6 | 1.2 | 1.0 | 11.0 |
| | CCAT | 1.00 | 0.0 | 100.0 | 0.0 | 1.0 | 6.7 |
| impulse_noise | Normal | 0.87 | 72.4 | 27.6 | 79.2 | 1.0 | 81.3 |
| | AT | 0.98 | 13.9 | 86.1 | 18.8 | 1.0 | 61.4 |
| | CCAT | 1.00 | 0.0 | 100.0 | 0.0 | 1.0 | 47.1 |
| motion_blur | Normal | 0.86 | 73.9 | 26.1 | 29.5 | 1.0 | 37.2 |
| | AT | 0.62 | 90.6 | 9.4 | 0.3 | 1.0 | 2.7 |
| | CCAT | 1.00 | 0.1 | 99.9 | 0.0 | 1.0 | 1.7 |
| rotate | Normal | 0.70 | 92.4 | 7.6 | 1.7 | 1.0 | 4.6 |
| | AT | 0.64 | 87.9 | 12.1 | 0.8 | 1.0 | 4.1 |
| | CCAT | 0.94 | 84.3 | 15.7 | 0.5 | 1.0 | 3.8 |
| scale | Normal | 0.84 | 89.5 | 10.5 | 0.7 | 1.0 | 3.1 |
| | AT | 0.86 | 78.5 | 21.5 | 0.1 | 1.0 | 3.0 |
| | CCAT | 0.85 | 91.0 | 9.0 | 0.3 | 1.0 | 2.3 |
| shear | Normal | 0.60 | 97.6 | 2.4 | 0.2 | 1.0 | 0.8 |
| | AT | 0.56 | 95.1 | 4.9 | 0.1 | 1.0 | 0.9 |
| | CCAT | 0.93 | 89.1 | 10.9 | 0.0 | 1.0 | 0.9 |
| shot_noise | Normal | 0.74 | 93.0 | 7.0 | 1.4 | 1.0 | 3.6 |
| | AT | 0.62 | 91.3 | 8.7 | 0.2 | 1.0 | 1.9 |
| | CCAT | 0.87 | 86.5 | 13.5 | 0.2 | 1.0 | 2.0 |
| spatter | Normal | 0.85 | 72.8 | 27.2 | 18.9 | 1.0 | 28.1 |
| | AT | 0.65 | 88.3 | 11.7 | 0.6 | 1.0 | 3.6 |
| | CCAT | 1.00 | 0.7 | 99.3 | 0.0 | 1.0 | 1.6 |
| stripe | Normal | 0.92 | 61.1 | 38.9 | 69.6 | 1.0 | 70.2 |
| | AT | 0.98 | 12.2 | 87.8 | 83.8 | 1.0 | 81.3 |
| | CCAT | 1.00 | 0.0 | 100.0 | 0.0 | 1.0 | 75.8 |
| translate | Normal | 0.72 | 95.2 | 4.8 | 0.3 | 1.0 | 1.6 |
| | AT | 0.80 | 74.6 | 25.4 | 0.2 | 1.0 | 4.0 |
| | CCAT | 0.86 | 91.7 | 8.3 | 0.1 | 1.0 | 1.3 |
| zigzag | Normal | 0.85 | 74.5 | 25.5 | 16.3 | 1.0 | 24.8 |
| | AT | 0.87 | 53.6 | 46.4 | 10.9 | 1.0 | 24.9 |
| | CCAT | 0.97 | 42.6 | 57.4 | 9.3 | 1.0 | 21.1 |

Table 13: **Per-corruptions results on MNIST-C.** *Results on MNIST-C, broken down by individual corruptions (first column);* `all` *includes all corruptions and* `mean` *are the averaged results over all corruptions. We report ROC AUC, FPR and additionally the true negative rate (TNR) in addition to the thresholded and unthresholded Err on the corrupted examples.*

| Corruption | Training | Detection Setting $\tau$@99%TPR | | | | | Standard Setting $\tau=0$ |
|---|---|---|---|---|---|---|---|
| | | ROC AUC | FPR in % | TNR in % | Err in % | $\tau$ | Err in % |
| | | | | | | | |

| Corruption | Training | ROC AUC | FPR in % | TNR in % | Err in % | $\tau$ | Err in % |
|---|---|---|---|---|---|---|---|
| all | Normal | 0.57 | 97.1 | 2.9 | 12.2 | 0.6 | 13.7 |
| | AT | 0.53 | 96.2 | 3.8 | 16.2 | 0.3 | 18.1 |
| | CCAT | 0.66 | 72.1 | 27.9 | 10.4 | 0.4 | 27.2 |
| mean | Normal | 0.57 | 97.1 | 2.9 | 12.3 | 0.6 | 13.7 |
| | AT | 0.53 | 96.2 | 3.8 | 16.2 | 0.3 | 18.1 |
| | CCAT | 0.66 | 72.1 | 27.9 | 8.5 | 0.4 | 27.2 |
| brightness | Normal | 0.50 | 98.1 | 1.9 | 7.5 | 0.6 | 8.4 |
| | AT | 0.50 | 97.0 | 3.0 | 14.9 | 0.3 | 16.5 |
| | CCAT | 0.54 | 94.8 | 5.2 | 8.2 | 0.4 | 10.4 |
| contrast | Normal | 0.52 | 98.1 | 1.9 | 8.4 | 0.6 | 9.4 |
| | AT | 0.60 | 94.1 | 5.9 | 17.1 | 0.3 | 20.0 |
| | CCAT | 0.55 | 96.6 | 3.4 | 10.3 | 0.4 | 11.9 |
| defocus_blur | Normal | 0.50 | 98.1 | 1.9 | 7.4 | 0.6 | 8.4 |
| | AT | 0.51 | 96.8 | 3.2 | 15.5 | 0.3 | 17.2 |
| | CCAT | 0.49 | 97.5 | 2.5 | 9.2 | 0.4 | 10.5 |
| elastic_transform | Normal | 0.60 | 97.3 | 2.7 | 12.1 | 0.6 | 13.5 |
| | AT | 0.57 | 95.8 | 4.2 | 19.0 | 0.3 | 21.1 |
| | CCAT | 0.54 | 96.5 | 3.5 | 13.2 | 0.4 | 14.9 |
| fog | Normal | 0.51 | 98.0 | 2.0 | 7.9 | 0.6 | 8.8 |
| | AT | 0.57 | 94.7 | 5.3 | 15.7 | 0.3 | 18.3 |
| | CCAT | 0.55 | 96.0 | 4.0 | 9.0 | 0.4 | 11.1 |
| frost | Normal | 0.57 | 97.2 | 2.8 | 11.5 | 0.6 | 12.8 |
| | AT | 0.53 | 95.9 | 4.1 | 16.0 | 0.3 | 18.0 |
| | CCAT | 0.65 | 88.1 | 11.9 | 9.0 | 0.4 | 12.5 |
| gaussian_blur | Normal | 0.50 | 98.0 | 2.0 | 7.4 | 0.6 | 8.4 |
| | AT | 0.51 | 96.8 | 3.2 | 15.5 | 0.3 | 17.1 |
| | CCAT | 0.49 | 97.4 | 2.6 | 9.2 | 0.4 | 10.4 |
| gaussian_noise | Normal | 0.62 | 96.2 | 3.8 | 15.7 | 0.6 | 17.7 |
| | AT | 0.51 | 96.8 | 3.2 | 15.5 | 0.3 | 17.0 |
| | CCAT | 1.00 | 0.0 | 100.0 | 0.0 | 0.4 | 84.9 |
| glass_blur | Normal | 0.76 | 92.5 | 7.5 | 40.8 | 0.6 | 43.1 |
| | AT | 0.58 | 95.4 | 4.6 | 18.0 | 0.3 | 20.2 |
| | CCAT | 0.93 | 31.1 | 68.9 | 14.7 | 0.4 | 31.8 |
| impulse_noise | Normal | 0.59 | 97.0 | 3.0 | 13.1 | 0.6 | 14.5 |
| | AT | 0.53 | 96.4 | 3.6 | 16.4 | 0.3 | 17.9 |
| | CCAT | 1.00 | 0.1 | 99.9 | 0.0 | 0.4 | 61.0 |
| jpeg_compression | Normal | 0.59 | 96.9 | 3.1 | 12.3 | 0.6 | 13.7 |
| | AT | 0.51 | 96.8 | 3.2 | 15.5 | 0.3 | 17.0 |
| | CCAT | 0.59 | 93.3 | 6.7 | 9.3 | 0.4 | 12.1 |
| motion_blur | Normal | 0.58 | 97.3 | 2.7 | 10.9 | 0.6 | 12.2 |
| | AT | 0.55 | 95.9 | 4.1 | 16.8 | 0.3 | 18.9 |
| | CCAT | 0.52 | 96.5 | 3.5 | 11.9 | 0.4 | 13.6 |
| pixelate | Normal | 0.54 | 97.6 | 2.4 | 9.7 | 0.6 | 10.9 |
| | AT | 0.51 | 96.7 | 3.3 | 15.3 | 0.3 | 17.0 |
| | CCAT | 0.52 | 97.0 | 3.0 | 9.2 | 0.4 | 10.8 |
| saturate | Normal | 0.55 | 97.5 | 2.5 | 10.0 | 0.6 | 11.3 |
| | AT | 0.55 | 95.3 | 4.7 | 18.2 | 0.3 | 20.5 |
| | CCAT | 0.48 | 97.6 | 2.4 | 11.8 | 0.4 | 13.0 |
| shot_noise | Normal | 0.58 | 97.0 | 3.0 | 12.1 | 0.6 | 13.7 |
| | AT | 0.51 | 97.1 | 2.9 | 15.4 | 0.3 | 16.7 |
| | CCAT | 1.00 | 0.0 | 100.0 | 0.0 | 0.4 | 84.5 |
| snow | Normal | 0.57 | 97.3 | 2.7 | 12.1 | 0.6 | 13.5 |
| | AT | 0.51 | 96.7 | 3.3 | 15.3 | 0.3 | 17.0 |
| | CCAT | 0.58 | 95.6 | 4.4 | 11.2 | 0.4 | 13.3 |
| spatter | Normal | 0.54 | 97.5 | 2.5 | 9.4 | 0.6 | 10.6 |
| | AT | 0.51 | 96.8 | 3.2 | 15.5 | 0.3 | 17.2 |
| | CCAT | 0.58 | 95.2 | 4.8 | 9.3 | 0.4 | 11.6 |
| speckle_noise | Normal | 0.57 | 97.0 | 3.0 | 12.4 | 0.6 | 13.8 |
| | AT | 0.51 | 97.1 | 2.9 | 15.2 | 0.3 | 16.7 |
| | CCAT | 1.00 | 0.0 | 100.0 | 0.0 | 0.4 | 83.0 |
| zoom_blur | Normal | 0.61 | 96.7 | 3.3 | 12.9 | 0.6 | 14.6 |
| | AT | 0.56 | 95.7 | 4.3 | 17.2 | 0.3 | 19.2 |
| | CCAT | 0.52 | 97.1 | 2.9 | 13.9 | 0.4 | 15.6 |

**CIFAR10:** Supplementary Results for Detection and Standard Settings ($L_\infty$ attack with $\epsilon = 0.03$ during training)

Table 14: ***Per-corruptions results on Cifar10-C.*** *Results on Cifar10-C focusing on individual corruptions (first column);* all *includes all corruptions and* mean *are the averaged results over all corruptions. We report ROC AUC, FPR and additionally the true negative rate (TNR) in addition to the thresholded and unthresholded Err on the corrupted examples.*

