# OpenReview forum: "Confidence-Calibrated Adversarial Training: Towards Robust Models Generalizing Beyond the Attack Used During Training"
_ICLR.cc/2020/Conference — Reject_

### Official Review · AnonReviewer1 · 2019-10-16
**Official Blind Review #1**

**Rating:** 6

**Review:**

*Edit after rebuttal*
See the comment below for a response to the author's rebuttal. I have updated my score to a weak accept.

Summary
========
This paper proposes a more granular adversarial training scheme, where the model is trained to have decaying confidence as the size of the adversarial perturbations increase.
The idea is quite natural and well motivated and presented. The paper suffers in its theoretical and evaluation sections though, which were both fairly confusing to me. I lean towards rejection for this paper but would consider changing my score if the presentation was improved.


Comments
=========
The observation that adversarial training enforces high confidence in the entirety of the epsilon ball is interesting, and the motivation for the proposed scheme is sound.

Yet, while the intuition of CCAT for addressing the accuracy-robustness trade-off makes sense to me, the purported beneficial effect for perturbations outside the given epsilon ball are not as clear.
The authors note in Section 3.1 that adversarial training gives no indications on how the model should behave outside the given epsilon ball, or on other threat models. While this is true, it is not clear how CCAT supposedly remedies this. By making the confidence decay within the epsilon ball, the training also doesn't explicitly enforce any behavior outside this ball or on other threat models. So if CCAT indeed provides benefits in those settings, it isn't clear why those gains should be expected.

The exposition of the technique in Section 3.2. is very clear, and Figure 1 nicely illustrates the effect of CCAT on the class confidences during an attack.

The theoretical toy example in Section 3.3. is a bit confusing, and I'm not convinced that the proof of Proposition 1 is correct. First, a classification problem parametrized by the class imbalance is a little weird in this setting. I have a few questions about the proof:
- You seem to be assuming that the model's confidence is calculated using a softmax. This assumption should be stated.
- The expressions for \hat{p}(y=2|x=x) and \hat{p}(y=1|x=x) are confusing to me. Why are you taking a log over the softmax? These probabilities are now negative... Also, in the second step, an equality of the form e^x / (e^y + e^x) = 1+e^(y-x) is used. This is incorrect, it should be 1/(1+e^(y-x)). In the end, the values log(1+e^a) and log(1+e^(-b)) are not in [0, 1] so they cannot be probabilities.
- From there on, it isn't clear what the rest of the proof is really saying

The experimental section is quite thorough and the results seem convincing. The considered adaptive attack, that optimizes (4) seems natural given that the defense thresholds on the model's confidence. I wonder how this is optimized though: Do you try a targeted attack for every possible target other than the true class?

I found the conflation of evaluation metrics quite confusing. It seems the authors know this and have done an effort to explain their choices, but I feel like this could still be improved further. For example, some of the metrics (e.g., AUC) should be maximized, while others (Err or RErr) should be minimized. This makes the reading of the result tables quite difficult.I would also suggest dropping one digit of precision in the Tables for better readability.

Overall, I have some mixed feelings about this paper. The idea is simple and well explained. The motivation, theoretical analysis, and reporting of results are somewhat confusing though. Some extra work on the presentation could definitely help this paper.

Some additional references
========================
- In the introduction, when mentioning the trade-off between robustness and accuracy, you should cite "Robustness May Be at Odds with Accuracy" by Tsipras et al. rather than Schmidt et al. (that paper shows that robustness may require more training data).
- Also in the introduction, you could consider referencing the following papers on robustness to different threat models:
    * Engstrom et al., "Exploring the Landscape of Spatial Robustness"
    * Tramer and Boneh, "Adversarial Training and Robustness for Multiple Perturbations"
- In the related work, you reference a number of works for adversarial training but not the original idea in the paper by Szegedy et al.
- In Section 3.1., the observation that the epsilon-ball around training examples can cross class boundaries is further analyzed in "Excessive Invariance Causes Adversarial Vulnerability" and "Exploiting Excessive Invariance caused by Norm-Bounded Adversarial Robustness" by Jacobsen et al.

Typos
=====
p.6 enumerator -> numerator

**Experience Assessment:**

I have published in this field for several years.

**Review Assessment: Checking Correctness Of Derivations And Theory:**

I carefully checked the derivations and theory.

**Review Assessment: Checking Correctness Of Experiments:**

I carefully checked the experiments.

**Review Assessment: Thoroughness In Paper Reading:**

I read the paper thoroughly.

---

> ### Author Response · Authors · 2019-11-14
> **Response to reviewer #1**
>
> We appreciate the comments by reviewer #1. In the revised version, changes marked in red, we improved the presentation of our experimental setup and results significantly.
>
> 1. How does CCAT motivate better model behavior beyond the epsilon-ball used during training?
>
> In contrast to AT, which encourages a constant, high-confidence prediction within the epsilon-ball, our CCAT encourages uniform prediction towards the border of the epsilon-ball and the transition into the uniform distribution is very flat for the power transition. Thus, the network is biased towards flat regions off the ``data generating distribution’’. As ReLU networks correspond to piecewise linear classifiers see e.g. [a], it can fit such a region very well. In contrast, AT enforces high confidence for the correct class inside the whole ball. If we consider the line connecting two samples of opposite classes then it is clear that the confidence has to change somewhere in between, and we have no control where this happens - thus adversarial examples of high confidence are unavoidable in normal AT once we leave the threat model during training. In our case we enforce uniform confidence between the classes so that we can detect these cases.
>
> [a] Hein et al., Why ReLU networks yield high-confidence predictions far away from the training data and how to mitigate the problem, CVPR’19
>
> 2. Correctness, assumptions and ambiguities in our theoretical result.
>
> We are grateful for noticing these typos which all have been fixed. The problem was that there was a -log missing in front of the expressions. The missing assumptions were hidden in the text before, but we agree that they should be put directly into the statement as in the revised version. Now, we also provide a detailed derivation of the expected loss of CCAT which is unfortunately a bit complicated as we have three probability distributions over the labels: the true one, the imposed one and the predicted one. We also added more details to the proofs for adversarial training. However, the result is correct as it was stated.
>
> 3. Optimization of our adaptive attack, i.e., Equation (4).
>
> No, we do not try K targeted attacks and take the highest-confidence one (where K is the number of classes). Instead, the maximum over classes (not including the true class) is directly optimized over - meaning we differentiate through the maximum explicitly. This is much more efficient and works also better than running 10 targeted attacks. The efficiency is especially important as we use the attack for training. Moreover, targeted attacks have the tendency to get more easily stuck in the transition to the regions where the predictions get
> uniform.
>
> 4. Evaluation metrics and presentation of results.
>
> As this point was raised by all reviewers, we include a detailed description of our experimental setup in a separate comment. In our revised version, we put significant effort into addressing the raised points by putting emphasis on our detection setting and differentiating it from standard evaluation, explaining our evaluation methodology and metrics in more detail and at more prominent places and simplifying the presented results. Changes were made throughout the whole paper, but specifically in the newly introduced Section 3 which summarizes our evaluation methodology.
>
> 5. Additional references.
>
> We thank the reviewer for the many additional references provided. We agree with reviewer #1 and the other reviewers that our related work section can be improved. Thus, in the revision we added a significantly longer discussion of related work, covering recent variants and insights regarding adversarial training and detection of adversarial examples (and evaluation of such methods). In this discussion as well as in the introduction and the main section, we included the references noted by the reviewer appropriately.

---

> > ### Author Response · Authors · 2019-11-15
> > **Follow-up regarding CCAT's behavior beyond the epsilon-ball**
> >
> > As a follow-up regarding point 1., i.e., how CCAT encourages better behavior beyond the epsilon-ball used during training, we now provide more discussion and illustrations in the appendix, Figure 9 and Appendix B.5. Specifically, on MNIST, for linearly interpolating between two test examples, we show that CCAT is indeed able to predict uniform distributions in between, even beyond the epsilon-balls used during training. Adversarial trianing (AT), in contrast, gives high-confidence predictions on nearly all interpolated examples. Thus, at one or multiple points, AT has to change the predicted class, resulting in radical changes regarding the predicted distributions, while the uniform prediction of CCAT can be extrapolated easily without any significant changes. These examples also illustrate how CCAT addresses the robustness-accuracy trade-off as discussed in our response to reviewer #3.

---

> > > ### Comment · AnonReviewer1 · 2019-11-15
> > > **Response to rebuttal**
> > >
> > > Thanks for the thorough rebuttal.
> > > One issue I still see is with your comment about the adaptive attack: you say that you don't do K targeted attacks because "the efficiency is especially important as we use the attack for training". While this may hold during training, the relative inefficiency of an attack shouldn't be used as a disqualifying criterion during the evaluation. Here, you should try the absolutely strongest attack you have. So I would encourage you to also add the K-targeted attack for evaluation.
> > >
> > > Overall, the rebuttal has addressed many of my concerns so I'll up my score to a weak accept.

---

### Official Review · AnonReviewer3 · 2019-10-17
**Official Blind Review #3**

**Rating:** 3

**Review:**

Summary:
This paper proposes the "confidence-calibrated adversarial training (CCAT)" to train robust DNNs against adversarial examples. The idea is to perform adversarial training with gradually smoothed network predictions (in an exponential or power rate to perturbation size). This new approach seems can help detection and defense at the same time to some extent. While the idea is quite interesting, the paper is poorly written and many times hard to read.

Here are some detailed comments:
1. Introduction. It is hard to get how the "robustness-accuracy trade-off" and the "poor generalization to other or stronger attacks" are addressed by CCAT. I can hardly agree that "standard adversarial training strongly depends on the training attack". PGD-AT generalizes pretty well to other types of attacks such as CW. It needs more concrete evidence to make such a claim.
2. Related work can be improved. There are a few places inaccurate descriptions, like "White-box attacks utilizing projected gradient ascent to ...". And since the focus of this paper is adversarial training, related works deserves more detailed discussions.
3. Section 3, the 100%/50% adversarial training is not properly explained. For example,  in (3), I cannot tell how the 50% comes from, as input x in the two components are the same.
4. The confidence calibration seems related to the reverse cross entropy (RCE) proposed in [*] for detection?
5. Figure 1, missing legends. I cannot find where the PGD-Conf is defined when came across Figures 1/2.
6. Experiments. The settings are not standard, which makes results hard to interpret. The Err, RErr metrics are confusing. Not sure what 99% TPR is for, detection?
7. Hard to tell the real improvement. For example , in Table 2, CIFAR-10, L_{\infty}=0.0.3, CCAT has even lower RErr (\tau =0) and ROC (\tau@99%TPR) than standard AT.

[*] Pang, Tianyu, et al. "Towards robust detection of adversarial examples."

Advances in Neural Information Processing Systems. 2018.
Missing citations  of a few recent papers in adversarial training:
[1] Zhang, Hongyang, et al. "Theoretically principled trade-off between robustness and accuracy." ICML, 2019.
[2] Wang, Yisen, et al. "On the Convergence and Robustness of Adversarial Training." ICML, 2019.
[3] Carmon, Yair, et al. "Unlabeled data improves adversarial robustness." NeurIPS, 2019.
[4] Uesato, et al. "Are Labels Required for Improving Adversarial Robustness?" NeurIPS, 2019.

A few additional high-level suggestions:
1) Show some concrete examples where standard adversarial training fails while CCAT is not.
2) Show the formulation of CCAT a little earlier in Section 3. Then link back to standard adversarial training or other defense methods. Not the other way around like it did now (taking too long to get to the idea of CCAT).
3. Use a standard experiment setting, and compare 1-2 more existing methods fairly.
4. Show understanding and analysis on CIFAR-10 dataset instead of SVHN.

=============
I appreciate the substantial revisions made during the rebuttal. The paper reads much better now. Unfortunately, my rating remains the same. My major concern goes to the unusually setting adopted in this paper. Detection and defense are two different types of adversarial research. I appreciate the efforts this paper has put to try combining them together into a unified robust defense (eg. detection first, then defense). Specifically, the use of confidence thresholding (equivalent to detection) before defense. This is also the main reason that causes the other reviewers hard to understand the real impact made in this paper. It is not fair to directly compare AT with CCAT, as AT does not have the detection component. I suggest the authors to include an existing detection method (LID (Ma et al. 2018), or Mahalanobis score for AT detection ("A Simple Unified Framework for Detecting Out-of-Distribution Samples and Adversarial Attacks"), before applying AT for prediction. Overall, the authors have done a good rebuttal, but it still requires much improvement to be accepted.

Typo:
1. Table 3, the 3rd column should be Cifar10 not SVHN.


**Experience Assessment:**

I have published one or two papers in this area.

**Review Assessment: Checking Correctness Of Derivations And Theory:**

I carefully checked the derivations and theory.

**Review Assessment: Checking Correctness Of Experiments:**

I carefully checked the experiments.

**Review Assessment: Thoroughness In Paper Reading:**

I read the paper thoroughly.

---

> ### Author Response · Authors · 2019-11-14
> **Response to reviewer #3**
>
> We thank reviewer #3 for the comments and suggestions. The corresponding changes in our revised paper are marked in red.
>
> 1a. Benefits of CCAT regarding the robustness-accuracy trade-off.
>
> The improved robustness-accuracy trade-off is motivated by our theoretical toy example in Proposition 1. It illustrates that there exist very simple problems where adversarial training is not able to be accurate and robust at the same time, while CCAT is able to. In practice, e.g., on Cifar10, this advantage can also be observed in the obtained test error. We hypothesize that the toy example of Proposition 1 might also occur on challenging datasets where, due to ambiguity between different classes, the epsilon-balls around training examples overlaps. This case cannot be handled by standard adversarial training, as it enforces constant high-confidence predictions on both epsilon-balls, even if they belong to different classes. CCAT, in contrast, can handle such cases better as the network is encouraged to have uniform prediction throughout the largest parts of both epsilon balls.
>
> 1b. “Standard adversarial training strongly depends on the training attack” not correct
>
> We agree that this statement was misleading. Even though in earlier times it was the case that weak attacks lead to weak defenses, e.g., using FSGM in adversarial training, this is not the case anymore. We actually meant that the performance of adversarial training is typically only good if the threat models at training and test time agree. When using larger epsilon-balls or different L_p norms (not a subset of the L_inf ball during training), adversarial training is not robust against these „unseen“ attacks (Table 3 in the revision).
>
> 2. Related work and missing citations.
>
> We thank the reviewer for the references. We improved our discussion of related work significantly, covering adversarial attacks, also going beyond epsilon-constrained ones, adversarial training and its variants and detection of adversarial examples.
>
> 3. 100% and 50%/50% adversarial training.
>
> In the revised version, we clarified 100% and 50%/50% adversarial training. In Equation (3), as we consider expectations over the data distribution, the balance between both determines the 50%/50% split. For example, when weighting the second, clean term, by a factor of two, this would correspond to training on ~66.6% clean examples and ~33.3% adversarial examples. This balance is, in practice, implemented by dividing the batch in each iteration.
>
> 4. Reverse cross-entropy training.
>
> We agree that [*] is similar to our approach. In the revised version, we include a discussion in our related work section and a detailed comparison of [*] with adversarial training and CCAT in Appendix B.6, Table 8. We note that the K-density detector [b] used has been shown to be ineffective against adaptive attacks [c], which is why we include the alternative non-maximum entropy detector in our experiments. We found that the approach of [*] is competitive on MNIST, but is not robust on SVHN and Cifar10.
>
> [b] Feinman et al., Detecting adversarial samples from artifacts, ArXiv, 2017.
> [c] Carlini and Wagner, Adversarial Examples Are Not Easily Detected: Bypassing Ten Detection Methods, ArXiv 2017.
>
> 5. Figures.
>
> We thank the reviewer for the notes regarding our figures. In the revised version we included legends and made sure that the captions are easier to understand.
>
> 6. Interpretation of evaluation metrics and significance of experimental results.
>
> As this point was also noted by the other reviewers, we include a detailed description of our experimental setup in a separate comment. We also put significant effort into revising the paper and making our experimental setup clearer.
>
> In the following we address the reviewer‘s specific points: Indeed, the setting is not standard as CCAT does not intend to correctly classify adversarial examples, but instead to detect them. In the revised version, we evaluate both adversarial training and CCAT only in our detection setting. This is also beneficial for adversarial training. While our metric is non-standard, by extending the well-known robust test error, our results are still comparable to related work. In our detection setting, CCAT can roughly match the robustness of adversarial training regarding the L_inf attack trained on, but improves significantly over adversarial training using larger perturbations or L_2, L_1 and L_0 attacks.
>
> 8. Concrete examples.
>
> In the revised version, we include and discuss concrete adversarial examples for adversarial training and CCAT, Figure 8 and Appendix B.5.
>
> 9. Structure.
>
> We thank the reviewer for the proposed high-level changes. While the revised version still introduces adversarial training first, we put more effort into having a clear outline and describing/motivating CCAT more clearly.
>
> 10. Analysis on Cifar10.
>
> We include analysis and ablation on Cifar10 in the appendix. We also added confidence plots (as in Figure 1) on Cifar10.

---

> > ### Author Response · Authors · 2019-11-15
> > **Follow-up regarding robustness-accuracy trade-off**
> >
> > To support our claims regarding point 1a, i.e., how CCAT addresses the robustness-accuracy trade-off, we now show additional results in the appendix, Figure 9 and Appendix B.5. In particular, on MNIST, we interpolate linearly between two test examples of different classes and show that CCAT resorts to uniform prediction between these two examples. Adversarial training (AT), in contrast, exhibits high-confidence predictions throughout the interpolation. Thus, CCAT is better able to handle cases where the epsilon-balls around training or test examples overlap. Additionally, it shows that CCAT is able to extrapolate uniform predictions beyond the epsilon-balls used during training, as discussed in our response to reviewer #1.

---

### Official Review · AnonReviewer2 · 2019-10-21
**Official Blind Review #2**

**Rating:** 3

**Review:**


====== AFTER READING THE AUTHOR RESPONSE ======

Many thanks for the extensive response and the respective revision from the author(s).
Mainly, I found the main results are adjusted in the revision to rather demonstrate its good detection performance at 99% TPR, and I feel the message of the manuscript becomes more strengthen. However, at the same time I feel the proposed method would be slightly less motivated without the improved results of the "pure" robust accuracy, as the method itself is anyway a variant of adversarial training. There are several works that specially focus on detecting adversarial examples [1, 2], and comparing the results with them on diverse threat models would more strengthen the paper.

In overall, I keep my score unchanged to the current version of manuscript.

[1] A Simple Unified Framework for Detecting Out-of-Distribution Samples and Adversarial Attacks, NeurIPS 2018
[2] The Odds are Odd: A Statistical Test for Detecting Adversarial Examples, ICML 2019

===============================================

The paper proposes a new adversarial training scheme to improve generalization of robustness over unseen threat models, e.g. larger perturbations or different noise distributions. The key idea is to impose uniform confidence on the adversarial examples depending on the distance from the original example, based on a pre-determined distance metric, e.g. L-infinity distance. Experimental results shows its effectiveness on detecting adversarial examples and unseen robustness for MNIST, SVHN, and CIFAR-10 datasets, under L-infinity and L-2 adversaries.

In overall, I agree that improving generalization over unforeseen adversaries is a very important problem in adversarial training, and the paper addresses this problem with a novel approach. The manuscript is generally well-presented with clear motivation. In particular, I appreciated the simplicity of the proposed idea, and the thoroughness of experiments as a defense paper, e.g. presenting per-example worst-case results across diverse attacks. However, I am currently on a slightly negative side, due to some unclear points in the experimental results. I would like to increase the score if the issue could be addressed, regarding the importance of the problem and their approach, which seems valuable to be shared in the community.

Mainly, it is still hard for me to interpret the presented experimental results at the positive side, unless the authors could further address on the important points of the results. In general, the results are not that clear as claimed, especially when tau=0, to show that CCAT improves robustness: At the original threat model (L-inf with the smallest epsilon), CCAT shows much inferior results across all the datasets. It does improves the L-2 results, except for CIFAR-10 which should be a bare baseline to show the scalability of the method. Although the paper also point out that CCAT sometimes achieves much lower clean test error, but sometimes this also signals the less robustness. I hope the paper could justify such points in the Table 2 for better presentation.

- Perhaps ResNet-20 is too small for CIFAR-10 tasks, as the training loss would hardly minimized into 0? I think WRN-10 is a more fair standard for CIFAR-10 in AT, and wonder if the result could show more effectiveness (or get more aligned tendency across datasets) of the proposed method if the capacity of network is increased.
- Apart from the thoroughness of the attack methods assumed, considering only L-inf and L-2 adversaries may be not enough to claim the "generalization ability" of a defense. Could the method also improves robustness against other attacks, e.g. general corruption (such as MNIST-C, CIFAR-10-C), or unrestricted adversarial examples?
- Eq 4: Does it mean that the logits other than the 2nd predictions are not considered when generating adversarial examples? Personally, I don't much get the motivation of using this objective, even while it is described in below Eq. 4.

**Experience Assessment:**

I have read many papers in this area.

**Review Assessment: Checking Correctness Of Derivations And Theory:**

I assessed the sensibility of the derivations and theory.

**Review Assessment: Checking Correctness Of Experiments:**

I assessed the sensibility of the experiments.

**Review Assessment: Thoroughness In Paper Reading:**

I read the paper at least twice and used my best judgement in assessing the paper.

---

> ### Author Response · Authors · 2019-11-14
> **Response to reviewer #2**
>
> We thank reviewer #2 for the comments and ideas regarding our experiments. We agree that our experimental evaluation as described in the submitted paper can be hard to understand. In the revised version we describe our evaluation methodology including the used evaluation metrics in more detail and at more prominent places; the changes are marked in red.
>
> 1. “Understanding and interpretation of evaluation metrics and experimental results.”
>
> As this point was raised by all reviewers, we include a detailed description of our experimental setup in a separate comment and put quite some effort into improving our manuscript. Specifically, we now present just the detection setting in the main paper and report the standard setting tau=0 in the appendix. This simplifies the presentation of the results significantly and highlights better the achievements of CCAT (please see Section 3 and 4).
>
> In the following we directly respond to the reviewer’s specific comments: Regarding the robust test error (RErr) for confidence threshold tau = 0 (standard setting), CCAT shows indeed inferior performance to adversarial training but this setting is not useful as for tau=0 the confidence of adversarial examples plays no role (just the change of the decision matters). However, CCAT can only work well in a detection setting, where adversarial examples are mainly filtered out via confidence thresholding. For confidence threshold tau@99%TPR (that means we discard at most 1% correctly classified examples), CCAT is able to match the robustness of AT on the threat model where it is trained on. On threat models unseen during training, CCAT is more robust than AT on Cifar10 and SVHN and partially on MNIST, again evaluated using tau@99%TPR. On Cifar10, CCAT is able to get significantly better test error and robust test error than AT in the detection setting.
>
> 2. Are better results with WRN-10 on Cifar10 possible?
>
> We conducted some preliminary experiments with a deeper architecture (ResNet-34 instead of ResNet-20) and both a wider and deeper architecture (WRN-10) as suggested by the reviewer. ResNet-34 is, unfortunately, not able to improve results so far; however, the model did not finish the same number of epochs as reported in the paper. Regarding WRN-10, training is even slower due to the significantly wider layers. This also impacts evaluation as we are running several attacks with many thousand iterations. So far, the results seem promising, although only roughly 50 epochs have been finished. For example, the (thresholded) robust test error (RErr) on larger L_inf perturbations (epsilon = 0.06) is reduced from 91% to 83% and it is possible that the results to improve significantly with the remaining 150 epochs.
>
> 3. Additional evaluation on the corrupted MNIST/Cifar10 datasets or other threat models.
>
> In the revision we added L_1 and L_0 attacks (in addition to the L_inf and L_2 attacks) where, again, CCAT outperforms AT on SVHN and CIFAR10 - for MNIST the results are mixed as AT also benefits significantly from the used confidence-thresholding, e.g., reducing robust test error (RErr) from 65.9% for tau=0 to 7.8% for tau@99%TPR. Furthermore, we evaluated performance on corrupted MNIST/Cifar10 as suggested. The main results are included in the new Table 5, as well as in Tables 13 and 14 at the end of the appendix of the revised version. We note that, again, the results need to be evaluated in our detection setting: Given a corrupted example, CCAT may reject it based on confidence. This can be understood as the model saying “I don’t know, the example might be corrupted too strongly”. If not rejected, CCAT should correctly classify the corrupted example. Thus, we report ROC AUC indicating how well CCAT can reject corrupted examples and the thresholded test error (Err) on the corrupted (non rejected) test examples. CCAT has a lower error rate on non-rejected examples than AT and normal training and thus is more reliable.
>
> 4. Motivation of our adaptive attack objective, Equation (3).
>
> (We also refer the reviewer to our response to reviewer #1 point 3.)
> The maximum over classes (except for the true class) is explicitly included in our optimization problem. This means that the other confidences (i.e., 3rd, 4th, 5th … highest confidence) matter: the “target” class, i.e., the class corresponding to the highest confidence apart from the true class, may change during optimization. The motivation of this objective is that we want to find high-confidence adversarial examples in “any other” class than the true class (this is the best to fool our detection scheme), resulting in an untargeted attack suited for adversarial training. As mentioned by reviewer #1, the alternative would be to pre-define the target class, and run the attack multiple times for all possible target classes. In practice, we found that this is not as effective and significantly less efficient.

---

### Public Comment · ~Florian_Tramer1 · 2019-09-27
**Comparison to multi-attack adversarial training**

Nice work!
It would be interesting to compare CCAT to a (more expensive) baseline that performs adversarial training for multiple perturbation types simultaneously, as we do in our upcoming NeurIPS paper: https://arxiv.org/abs/1904.13000
It would be great if CCAT could achieve comparable robustness at a lower cost.

---

> ### Author Response · Authors · 2019-10-18
> **Thanks for the Interesting Baseline**
>
> Thank you for bringing up this baseline and sorry for the late response. It seems quite interesting to compare against this type of adversarial training on different perturbation types - in our case $L_\infty$ and $L_2$ using different $\epsilon$-balls. We will conduct some basic experiments and report back.

---

> ### Author Response · Authors · 2019-11-14
> **Preliminary experiments with multi-attack adversarial training**
>
> We are sorry for the late response, however, we had some difficulties training the max-variant of your proposed method in our setting, i.e., using L_inf and L_2 attacks using the epsilon-balls reported in the paper. We found that, during training, the L_inf attacks are usually “stronger”, such that the maximum focuses only on those, neglecting L_2 adversarial examples. This results in poor robustness against L_2 attacks at test time. While we tried to adapt the used epsilon-balls (in order to balance the L_inf and L_2 attack) and adapt the number of iterations (as done in your paper), we resorted to the average-variant instead. In preliminary results on MNIST, after roughly 30 epochs with 20 iterations for the L_inf attack and 40 iterations for the L_2 attack, robustness approaches the robustness of standard adversarial training and our CCAT, with roughly 6% robust test error (RErr) after thresholding, compared to 1% for adversarial training and 7% for CCAT. On larger L_inf perturbations, however, RErr increases again to roughly 97%. On L_2, in contrast, robustness improves from 81% (adversarial training) to 35%, compared to 1% for CCAT. On Cifar10, considering 50 epochs, results seem to point in a similar direction. Robustness against the L_2 attack is improved marginally from 73% (adversarial training) to 67% compared to 46% for CCAT. However, we expect the results to improve in the remaining 50 epochs.

---

### Author Response · Authors · 2019-11-14
**Explanation of experimental setup and evaluation metrics**

In the following, we briefly clarify our experimental setup as questions regarding our evaluation were raised by all three reviewers.

While adversarial training intents to correctly classify adversarial examples, CCAT runs in a two-stage process: First, (potentially adversarial) examples can be rejected by confidence thresholding. Here, ideally, CCAT rejects all adversarial examples as it encourages low-confidence adversarial examples, while not rejecting any correctly classified (clean) examples. Second, robustness and accuracy are evaluated on the non-rejected (i.e., correctly or incorrectly classified) examples. Here, if CCAT does not reject an example, it should be classified correctly, irrespective of whether it is an adversarial or a clean example.

These two stages are evaluated separately. For the first stage, we consider successful adversarial examples as negatives and correctly classified test examples as positives. Then, ROC AUC (higher is better) quantifies how well CCAT can discriminate between adversarial examples and correctly classified (clean) examples. For the second stage, after confidence thresholding, we fix the threshold tau corresponding to a true positive rate (TPR) of 99%; this means that at least 99% of correctly classified examples have to be accepted (not rejected), or equivalently at most 1% of correctly classified test examples can be rejected.

On the non-rejected examples, we measure accuracy and robustness using test error (Err) and robust test error (RErr). However, extending these metrics to our confidence-thresholded setting is non-trivial. For example, correctly classified examples can can have lower confidence than their corresponding adversarial examples. In practice, this raises some difficulties. In our formulation of the confidence-thresholded RErr, we take these special cases into account in order to obtain a formulation that (a) reduces to the standard (non-thresholded) robust test error used in related work for confidence threshold tau=0 and (b) is constrained to [0, 1] and, thus, easily interpretable. As result, our confidence-thresholded RErr is fully comparable to the standard RErr reported in the literature.

Finally, we note that confidence-thresholding is also beneficial for standard adversarial training. Both Err and RErr can only improve when employing confidence-thresholding, as we show in the paper for regular adversarial training on all datasets. Thus, it is not only meaningful but also in favor of our adversarial training baseline to compare to CCAT in our two-stage detection setting.

---

### Decision · Program_Chairs · 2019-12-19

**Decision:**

Reject

**Comment:**

This paper proposes a confidence-calibrated adversarial training (CCAT). The key idea is to enforce that the confidence on adversarial examples decays with their distance to the attacked examples. The authors show that CCAT can achieve better natural accuracy and robustness. After the author response and reviewer discussion, all the reviewers still think more work (e.g., improving the motivation to better position this work, conducting a fair comparison with adversarial training which does not have adversarial example detection component) needs to be done to make it a strong case. Therefore, I recommend reject.